# Plasma-derived extracellular vesicles from *Plasmodium vivax* patients signal spleen fibroblasts via NF-kB facilitating parasite cytoadherence

Haruka Toda et al.[#]

*Plasmodium vivax* is the most widely distributed human malaria parasite. Previous studies have shown that circulating microparticles during *P. vivax* acute attacks are indirectly associated with severity. Extracellular vesicles (EVs) are therefore major components of circulating plasma holding insights into pathological processes. Here, we demonstrate that plasma-derived EVs from *Plasmodium vivax* patients (*Pv*EVs) are preferentially uptaken by human spleen fibroblasts (*h*SFs) as compared to the uptake of EVs from healthy individuals. Moreover, this uptake induces specific upregulation of ICAM-1 associated with the translocation of NF-kB to the nucleus. After this uptake, *P. vivax*-infected reticulocytes obtained from patients show specific adhesion properties to *h*SFs, reversed by inhibiting NF-kB translocation to the nucleus. Together, these data provide physiological EV-based insights into the mechanisms of human malaria pathology and support the existence of *P. vivax*-adherent parasite subpopulations in the microvasculature of the human spleen.

[#]A list of authors and their affiliations appears at the end of the paper.

Extracellular vesicles (EVs) are a diverse group of cell-derived membrane structures secreted by living cells, present in biological fluids including plasma, and holding insights into pathological processes. EVs are attracting scientific attention due to their role in intercellular communication and their translational potential as therapeutic agents and non-invasive biomarkers of disease[1]. As intercellular communicators, compelling evidence is developing on their role in a wide range of pathologies, including parasitic diseases[1,2]. In the particular case of falciparum malaria, the most devastating parasitic disease, circulating EVs were correlated with cerebral malaria disease severity[3]. Moreover, EVs from in vitro culture of infected red blood cells (iRBCs) induced the formation of gametocytes, the sexual stages responsible for malaria transmission[4,5]. Other studies showed that EVs contained nucleic acids capable of triggering signalling mechanisms involved in parasite virulence[6] in modulating endothelial cells barrier properties[7] and in modulating immune responses[8,9]. These results demonstrate that EVs from in vitro culture of *P. falciparum* or from experimental infections in rodent malaria models, carry parasite-cargo acting as intercellular communicators. However, a direct link of circulating EVs from natural human malaria parasite infections and mechanistic insights into pathology has yet to be reported.

Pathology in human malaria caused by *P. falciparum* is mainly attributed to the ability of maturing parasite iRBCs to express adhesive variant proteins collectively called erythrocyte membrane protein-1 at their surface and cytoadhere to endothelial receptors, notoriously the intercellular adhesion molecule (ICAM-1), in the microvasculature of different organs[10]. In addition, *P. falciparum*-iRBCs and their secreted products induce ICAM-1 expression on brain endothelium linked to nuclear translocation of the NF-kB transcription factor[11]. These studies have been facilitated by the in vitro culture system of *P. falciparum*[12], from which an unlimited source of parasite iRBCs can be obtained for adhesion experiments. In the case of *P. vivax*, the most widely distributed human malaria parasite, a robust in vitro culture system is not available[13]. Nevertheless, adherence of patient-derived *P. vivax* iRBCs to endothelial cells and placental glycosaminoglycans has been demonstrated[14–16]. Moreover, *P. vivax* iRBCs express variant surface proteins called VIR[17], a member of which has been shown to bind to the ICAM-1 receptor[18]. In addition, in vivo studies using a reticulocyte-prone, non-lethal rodent malaria model, resembling *P. vivax* in these biological aspects, demonstrated active cytoadherence of iRBCs to spleen barrier cells of fibroblastic origin in the red pulp[19]. Furthermore, indications of iRBC accumulation in the spleen during *P. vivax* infections were observed in a clinical case of a *P. vivax* patient involving a ruptured spleen[20], and in a tissue pathology study involving a monkey experimentally infected with *P. vivax*[21]. These studies further support the assumptions of sequestration[22] or concealment of *P. vivax* iRBCs in the spleen[23,24].

Here, we hypothesize that circulating EVs in the plasma of *P. vivax* patients (*Pv*EVs) contain parasite proteins and signal human spleen fibroblasts facilitating the adhesion of infected reticulocytes. Our results show that plasma-derived EVs from patients contain parasite proteins and are specifically uptaken by human spleen fibroblasts (*h*SFs) inducing upregulation of ICAM-1 linked to nuclear translocation of NF-kB. Upon uptake of *Pv*EVs by *h*SFs, *P. vivax*-infected reticulocytes from patients show specific adhesion properties reversed by inhibiting NF-kB translocation to the nucleus. Together, these data provide physiological EV-based insights into the mechanisms of human malaria pathology and support the existence of *P. vivax*-adherent parasite subpopulations in the microvasculature of the human spleen.

## Results

**Isolation and characterization of plasma-derived EVs**. To initiate these studies, we used size-exclusion chromatography (SEC) to isolate EVs from individual patients and healthy donors, as this single technology purifies EVs and removes abundant soluble plasma proteins[25]. Moreover, we had previously shown that SEC fractions 7, 8 and 9 contained the plasma EVs-enriched fractions[25–27]. Thus, we isolated plasma-derived EVs from ten *P. vivax* patients from Brazil (Supplementary Data 1) and from ten healthy donors from Spain and obtained individual pools of these SEC fractions from each subject. We measured their protein concentration and showed a large variability among the different samples (Fig. 1a); yet, protein concentration was significantly enriched in EVs-enriched SEC fractions from patients compared to healthy individuals (Mann–Whitney test *$p = 0.0232$). Next, we aimed to identify *P. vivax* proteins associated with EVs from infections. We performed mass spectrometry based proteomic analysis of the ten individual samples from *P. vivax* patients and of the ten individual samples from healthy donors as controls. Overall, we identified 533 human proteins and 20 parasite proteins (Supplementary Data 2 and 3). On average, EVs from healthy donors contain a higher number of human proteins than EVs from infection suggesting a selective sorting of the human cargo on EVs during vivax infections (Supplementary Fig. 1). As expected, proteins belonging to exosomes, extracellular space, plasma membrane and extracellular region were highly enriched in GO analysis (Supplementary Fig. 1 and Supplementary Data 4). Of note, three parasite proteins were detected in EVs from infections with different unique peptides, the merozoite surface protein 3 (MSP3.1), the *Plasmodium* exported protein (PHISTc) and the glyceraldehyde-3-phostate dehydrogenase (GAPDH) (Fig. 1b). Of these three proteins, available antibodies against two, MSP3.1 and PHISTc, confirmed their association with *Pv*EVs by western blot analysis (Fig. 1c). As previously described, we observed MSP3.1[28] and PHISTc[29] bands of different molecular sizes. Together, these data demonstrate that circulating EVs from patients during acute attacks of vivax malaria are in higher concentration when compared to EVs from healthy donors and contain parasite-protein cargo.

Due to the limited amount of peripheral blood ethically approved to be withdrawn from patients during acute attacks, plasma-derived EVs (SEC fractions 7, 8 and 9) from the ten individual patients were pooled (*Pv*EVs) to pursue functional studies. We also made a single pool of these same SEC fractions from circulating EVs of the ten healthy donors (*h*EVs). Quantification of these pools by nanoparticle tracking analysis (NTA) showed that, in spite of their intrinsic variability in terms of protein concentrations, plasma from *P. vivax* patients still contained twice the concentration of EVs compared to healthy donors (Fig. 1d). In addition, beads-based flow cytometry using known EV markers (CD9, CD63, CD81, GAL3, CD5L and CD71) demonstrated similar median fluorescence intensity (MFI) values in pair-wise comparisons between *Pv*EVs and *h*EVs, except for CD9, CD63, CD81 and CD71, which had significantly higher MFI values in the *Pv*EVs (unpaired *t*-test, ****$p < 0.0001$, **$p = 0.0032$, ***$p = 0.0006$, ***$p = 0.0006$, respectively) (Fig. 1e). Of note, the transferrin receptor (CD71) had the highest MFI values which are consistent with the fact that CD71 is one of the most abundant proteins identified in proteomics analysis of reticulocyte-derived EVs[27].

**In vivo distribution**. To determine the in vivo distribution of EVs that may be associated with *P. vivax* infections, fluorescently labelled EVs were injected into immunocompetent mice via the

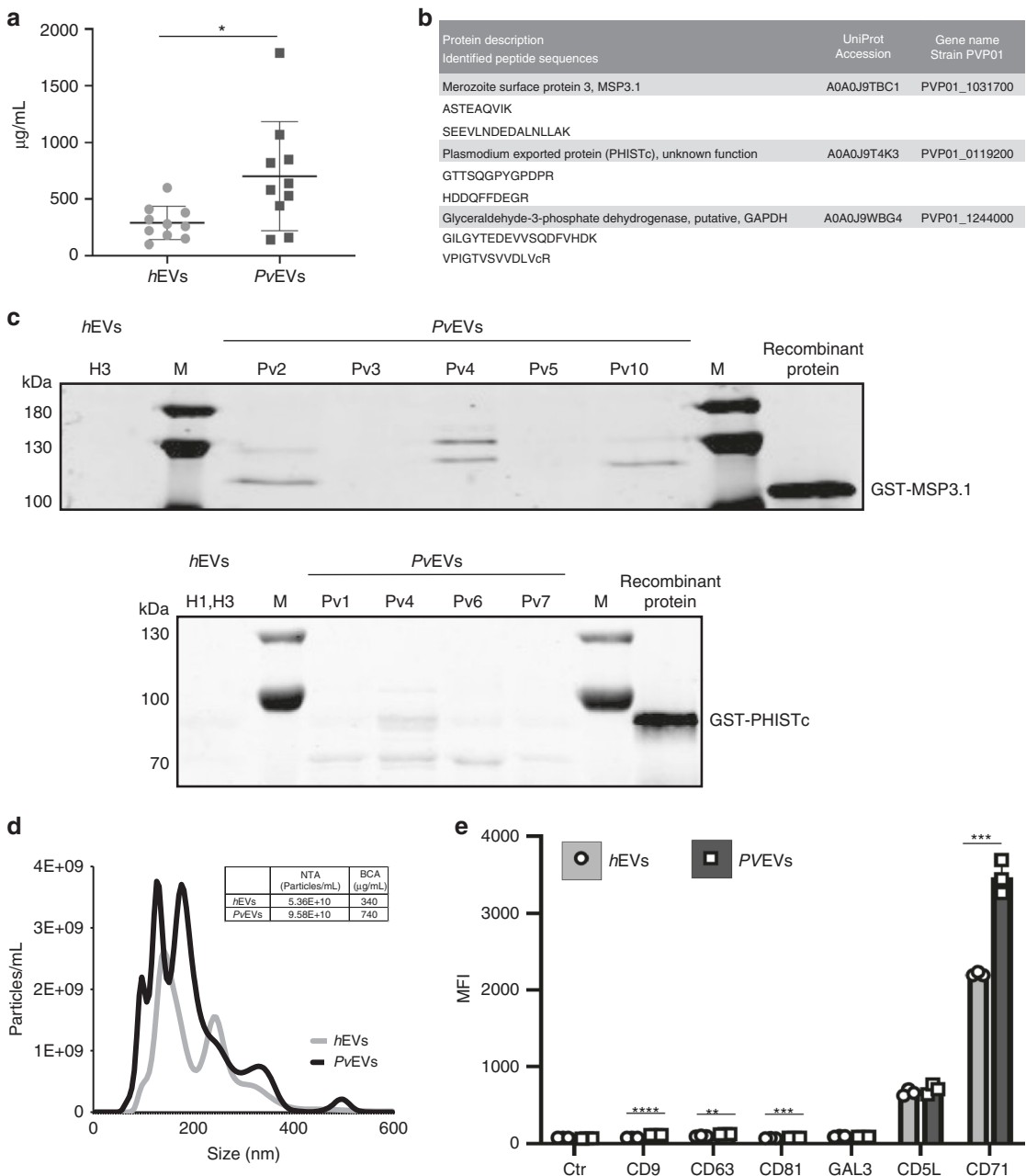

**Fig. 1 Characterization of plasma-derived extracellular vesicles from *P. vivax* patients. a** BCA (protein concentration) of circulating EVs from healthy donors (*h*EVs) and *P. vivax* patients (*Pv*EVs) (*n* = 10, individual samples). Data show mean ± SD. Two-sided, Mann–Whitney test, *\*p* = 0.0232 (GraphPad). **b** *P. vivax* proteins identified by different unique peptides (sequences below the description of corresponding proteins). UniProtKB accession numbers and gene name corresponding to the *P. vivax* PvP01 strain are shown. **c** Western blot analysis of *Pv*EVs obtained from different individual patients (Pv1-7 and Pv10) and human donors (H1 and H3) using anti *P. vivax* MSP3.1 (upper membrane) and PHISTc (bottom membrane) antibodies. MSP3.1 and PHISTc recombinant truncated-proteins fused to GST, were used as positive controls. Molecular weight in kDaltons (kDa) is shown to the left. M: molecular size marker. Image representative of three independent experiments. **d** Nanoparticle tracking analysis (NTA) profile (size [nm] versus concentration [particles/mL]) of pooled *h*EVs and *Pv*EVs was analysed using NanoSight LM10-12. Table shows mean of three measurement of pooled *h*EVs and *Pv*EVs quantified by NTA (particles concentration) and BCA (protein concentration). **e** Beads-based flow cytometry analysis of pooled *h*EVs and *Pv*EVs using six EV markers (CD9, CD63, CD81, GAL3, CD5L and CD71). Data show median fluorescence intensity (MFI) of each antibody and control antibodies (rabbit and mouse-isotype) ± SD (technical replicates, *n* = 3). Unpaired and two-sided, *t*-test *\*\*p* = 0.0032 (CD63), *\*\*\*p* = 0.0006 (CD81, CD71), *\*\*\*\*p* < 0.0001 (CD9) (GraphPad). Source data are provided as a Source data file and Supplementary Data 2, 3 and 4.

retro-orbital venous sinus, and the intensity of fluorescence in each organ was quantified after 1-h exposure (Fig. 2a). In spite of injecting the same quantity of labelled EVs, we observed a greater than three-fold increase in the uptake of *Pv*EVs versus *h*EVs and *Fasciola hepatica* EVs (*Fh*EVs) in the spleen and the liver (Fig. 2a). As immunocompetent mice might alter the in vivo

distribution of EVs from different species, we performed similar experiments in an immunosuppressed animal and observed the same trend in distribution (Supplementary Fig. 2a,b). In addition, we also observed similar tropisms using reticulocyte-derived EVs from BALB/c mice infected with the reticulocyte-prone *P. yoelii* strain (Supplementary Fig. 2c).

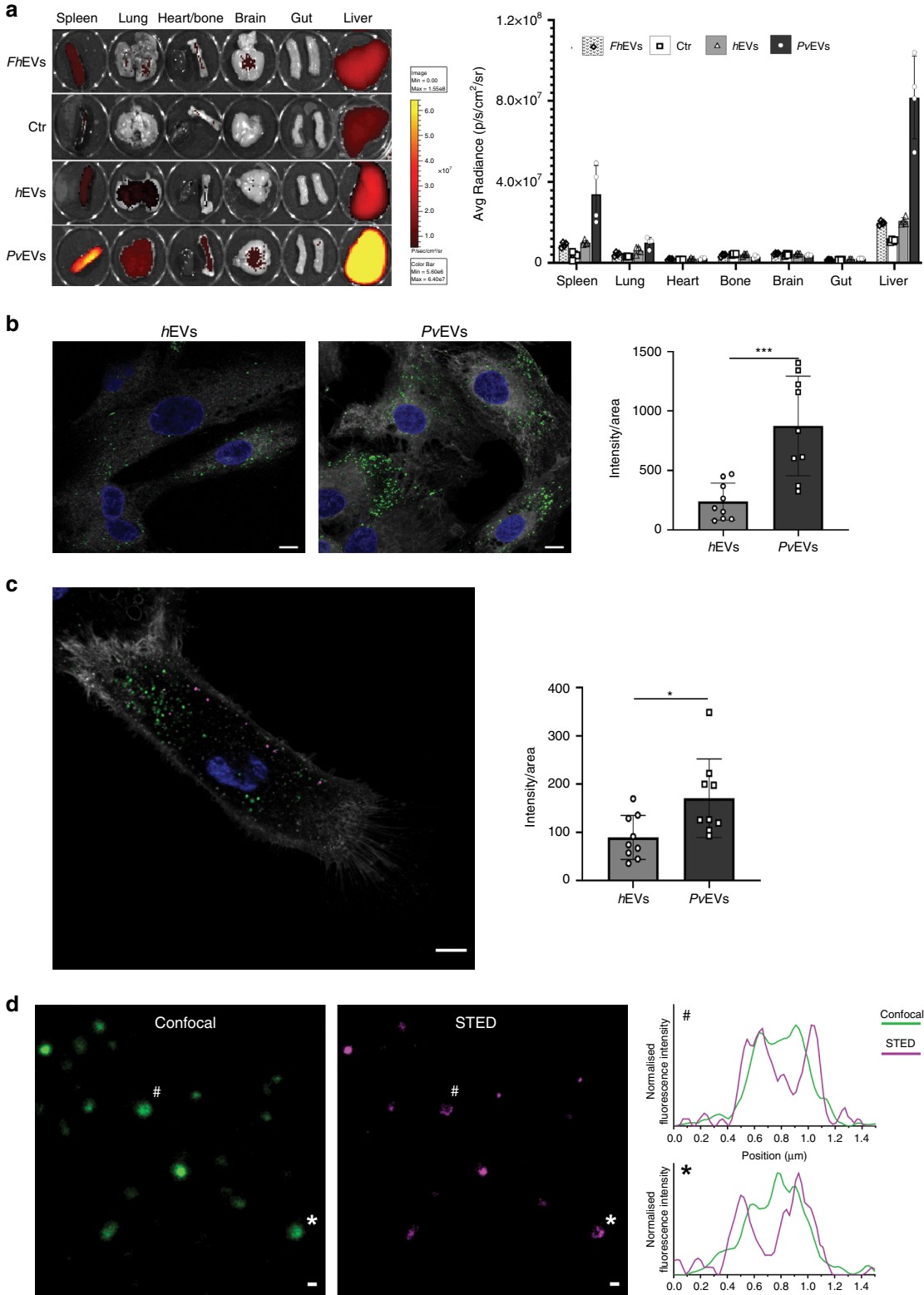

**Uptake of *Pv*EVs by human spleen fibroblasts**. As we hypothesized that EVs circulating in plasma from patients with *P. vivax* signal human spleen fibroblasts (*h*SFs) facilitating the adhesion of *P. vivax*-infected reticulocytes, we tested whether *Pv*EVs specifically interact with these cells. After ethical clearance, we obtained *h*SFs from transplantation donors and showed that primary *h*SFs displayed the typical morphology and markers of fibroblasts in in vitro culture (Supplementary Fig. 3). Next, we labelled independently *Pv*EVs and *h*EVs with PKH67, a lipophilic membrane fluorescence dye. Unbound dye was removed after washing five times through Amicon 100-kDa cut-off filters (Supplementary Fig. 4); thus, avoiding this confounding factor. Both preparations were incubated with *h*SFs for 2 h, the peak of active uptake (Supplementary Fig. 5) and images were randomly captured and quantified. Active uptake was further demonstrated by performing experiments at 4 °C and in fixed *h*SFs

**Fig. 2 In vivo distribution of EVs and taken up experiments. a** Representative IVIS-Spectrum images (left) and quantification of EVs signal to each organ (right). Three micrograms of Burgundy-labelled pooled EVs (*Fh*EVs, *h*EVs and *Pv*EVs), quantified by BCA and corresponding, respectively, to 9.08E + 9, 1.50E + 10 and 1.38E + 10 particles as measured by NTA, and Burgundy-labelled media as control, were injected retro-orbitally into C57Bl mice ($n = 4$, individual animals). After 1 h, mice were sacrificed to analyse several organs as indicated. Data show mean ± SD. **b, c** Uptake experiments. Confocal images (left) and quantification (right) of EVs uptaken by each cell ($n = 9$ cells examined over one independent experiment) after 2 h incubation. Data representative of three independent experiments. Nucleus was stained with Hoechst® 33342 (blue), and cell membrane with Cell Mask™ (white). Scale bar: 10 µm. **b** Pooled *h*EVs and *Pv*EVs were individually stained with PKH67 (green). Data show mean ± SD. Two-sided, Mann–Whitney test, **$p = 0.0008$ (GraphPad). **c** Pooled *h*EVs and *Pv*EVs were stained with PKH26 (magenta) and PKH67 (green), respectively, and competitively uptaken by *h*SFs. Data show mean ± SD. Two-sided, Mann–Whitney test, *$p = 0.0244$ (GraphPad). **d** Abberior STAR 580 stained *Pv*EVs were incubated with *h*SFs for 2 h at 37 °C. Confocal (green) and STED (magenta) images of labelled EVs membranes (left), and line profiles of normalised fluorescence intensities of EVs indicated by # and * (right). Scale bar: 400 nm. Image representative of three independent experiments. Source data are provided as a Source data file.

(Supplementary Fig. 5). Results demonstrated that significantly more *Pv*EVs were uptake by *h*SFs than *h*EVs (Mann–Whitney test ***$p = 0.0008$) (Fig. 2b). As EVs from non-infected cells and EVs from *P. vivax*-infected reticulocytes circulate together during natural infections, we performed competition uptake experiments by labelling *h*EVs with PKH26 (magenta) and *Pv*EVs with PKH67 (green). Results demonstrated that under these competing situations, *Pv*EVs were still preferentially uptake by *h*SFs as compared with *h*EVs albeit at lower levels as evidenced by the significance observed (Mann–Whitney test *$p = 0.0244$) (Fig. 2c). Of note, we always observed spots larger than 300 nm in the images used for quantifications. Therefore, we used STED super-resolution microscopy to determine if larger confocal spots corresponded to several EVs tethered together. Increased resolution imaging yielded ring-like structures corresponding to two-dimensional projections of a single spherical structure (Fig. 2d). This indicated that imaged >300 nm EVs represent individual vesicles rather than multi-vesicle aggregates. Since bigger confocal signals seem to correspond to a single entity, all further quantification experiments treated these signals as such.

**Specific upregulation of ICAM-1 in *h*SFs**. Since ICAM-1 can be a major player in malaria pathology[10], and it acts as a receptor for specific binding of a vivax variant surface protein, VIR14[18], we tested its expression in *h*SFs by qRT-PCR upon uptake of *Pv*EVs. We also tested the expression of genes coding for *vascular endothelium growth factor A* (*VEGFa*), *granulocyte-macrophage colony-stimulating factor* (*GM-CSF*), interleukins (*IL-6*, *IL-10*), toll-like receptors (*TLR4*, *TLR7*, *TLR9*), the chemokines (*CCL2*, *CXCL12*) and for the *alpha-actin-2* (*ACTA2*), as they have been shown to be expressed in human fibroblastic reticular cells in the spleen[30]. Lastly, we tested for the expression of *fibroblast growth factor 8* (*FGF8*), as experimental infections of mice with *P. yoelii* showed that it was upregulated in spleen fibroblasts[19]. As endogenous gene, *guanine nucleotide-binding protein subunit beta-2-like 1* (*GNB2L1*) was also included for qRT-PCR analysis.

Expression of *ICAM-1* was significantly upregulated at the transcript level (unpaired *t*-test *$p = 0.0473$) (Fig. 3a), whereas no significant differences in expression were observed for the other genes. Notably, dose-dependent upregulation of ICAM-1 was also observed at the protein level [unpaired *t*-test ***$p = 0.0005$ (20 µg) and ****$p < 0.0001$ (60 µg)] (Fig. 3b).

ICAM-1 is a transmembrane protein that plays pivotal roles in inflammatory, immunological and pathological processes and its expression can be upregulated by tumour necrosis factor-α (TNF-α), a cytokine present during malaria attacks and specifically release during *P. vivax* paroxysms[31]. To exclude this confounding factor, we confirmed the absence of TNF-α in the proteomic analysis of *Pv*EVs (Supplementary Data 2). Next, we determined ICAM-1 protein levels of *h*SFs activated by lipopolysaccharide (LPS) and TNF-α. Remarkably, only LPS induced ICAM-1 expression in these cells (Fig. 3b). These results thus provide

strong evidence that *Pv*EVs specifically signal *h*SFs for expression of the ICAM-1 receptor independently of TNF-α.

**ICAM-1 upregulation is linked to NF-kB nuclear translocation**. ICAM-1 expression is upregulated via NF-kB in *P. falciparum* in brain endothelium[11]. We therefore determined the subcellular location of NF-kB in *h*SFs after activation by *Pv*EVs. NF-kB remained in the cytoplasm in cells treated with *h*EVs, whereas nuclear translocation was specifically observed in cells exposed to *Pv*EVs (Fig. 3c). Treatment with a specific inhibitor of NF-kB, Bay11-7082, blocked the nuclear translocation of NF-kB mediated by *Pv*EVs (Fig. 3c). To quantify this translocation, we developed an image method by calculating NF-kB translocation ratio, dividing integrated density (IntD) of nuclear by IntD of perinuclear ($IntD_{Nuclear}/IntD_{Perinuclear}$). Results showed a highly significant difference between *h*SFs taken up by *Pv*EVs as opposed to *h*EVs (unpaired *t*-test ****$p < 0.0001$) (Supplementary Fig. 6).

To determine the association of NF-kB nuclear translocation with ICAM-1 expression, cells were incubated with either *Pv*EVs or *h*EVs in the presence or absence of Bay11-7082 and ICAM-1, and surface expression on *h*SFs was determined by flow cytometry. As shown in Fig. 3d, expression of surface ICAM-1 in cells that had uptaken *Pv*EVs was inhibited by treatment with 5 µM Bay11-7082. We also determined the subcellular localization of NF-kB in human spleen endothelial cells as they also likely uptake *Pv*EVs in the microvasculature of the spleen. In contrast to *h*SFs, nuclear translocation of NF-kB was readily observed in human spleen endothelial cells grown in culture medium as well as in cells incubated with *h*EVs or *Pv*EVs. Notably, the presence of NF-kB in the nuclei of these cells did not increase ICAM-1 expression (Supplementary Fig. 7).

**Cytoadherence of *P. vivax*-infected reticulocytes to *h*SFs**. To determine if the uptake of *Pv*EVs by *h*SFs could have a role in cytoadherence of *P. vivax* iRBCs, we performed in vitro binding experiments using parasite iRBCs isolated from the blood of *P. vivax* patients. Of note, in addition to the SEC pool of *P. vivax* patients from Brazil, we also included another SEC pool of three *P. vivax* patients from Colombia (Supplementary Data 1). Moreover, to mimic the concentrations of circulating EVs in both *P. vivax* patients and healthy donors, we used equal volumes of purified plasma EVs for these functional assays. Results demonstrated that binding to *h*SFs by *P. vivax* iRBCs is significantly higher in cells that had uptake *Pv*EVs, as opposed to cells that had uptake *h*EVs (Mann–Whitney test *$p = 0.0286$) (Fig. 4a). Worth of mentioning, mature multinucleated parasites were predominantly observed (Supplementary Fig. 8). Notably, such binding was inhibited by treatment of the cells with Bay11-7082 which inhibits nuclear translocation of NF-kB and surface expression of ICAM-1 in *h*SFs (Fig. 3c, d).

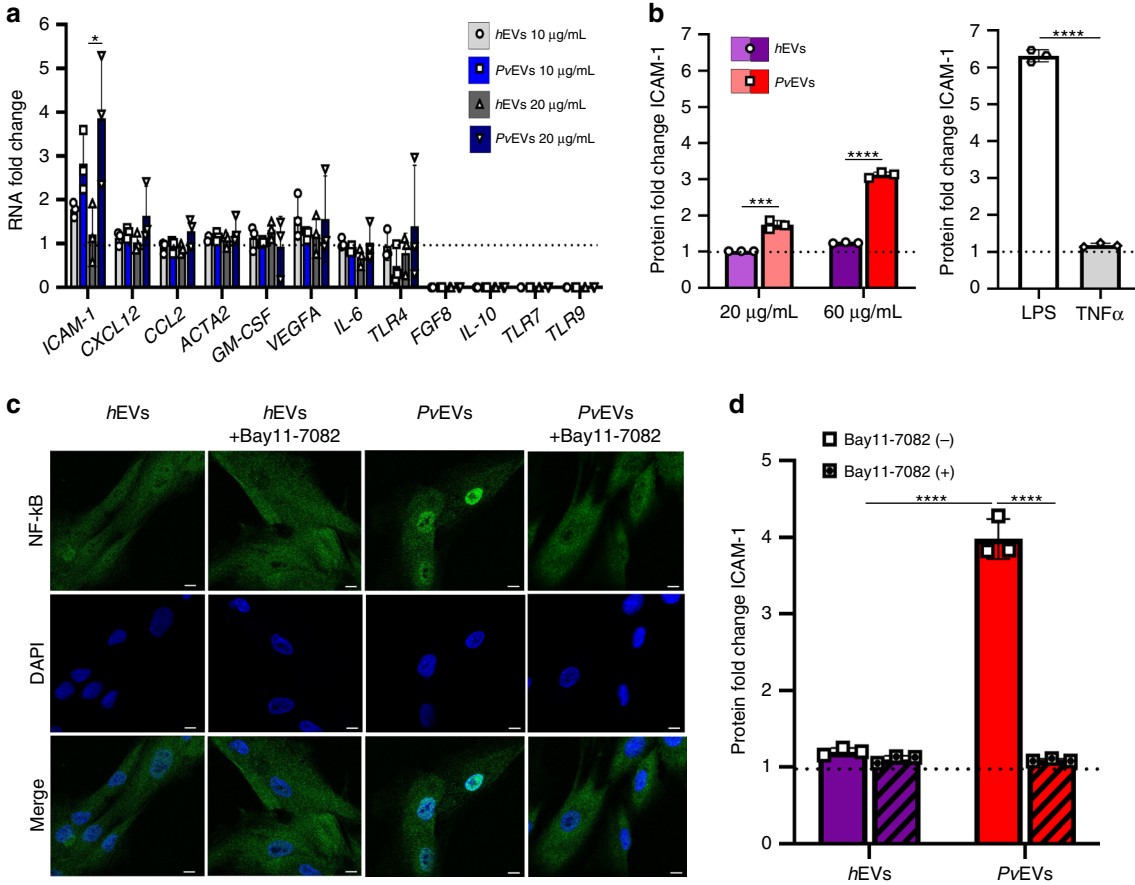

**Fig. 3 ICAM-1 expression and NF-kB nuclear translocation in *h*SFs cells. a** RNA expression. *h*SFs were incubated with media alone (Ctr) or with pooled *h*EVs or *Pv*EVs for 24 h. Quantitative RT-PCR was performed to analyse the expression of selected genes (*intercellular adhesion molecule 1 (ICAM-1)*, *C-X-C motif chemokine ligand 12 (CXCL12)*, *C-C motif chemokine 2 precursor (CCL2)*, *alpha-actin-2 (ACTA2)*, *granulocyte-macrophage colony-stimulating factor (GM-CSF)*, *vascular endothelial growth factor A (VEGFa)*, *interleukin-6 (IL-6)*, *Toll-like receptor 4 (TLR4)*, *fibroblast growth factor 8 (FGF8)*, *interleukin-10 (IL-10)*, *Toll-like receptors 7 and 9 (TLR7, TLR9)*) in *h*SFs. Data show mean of the fold-change relative to control values ± SD (technical replicates, $n = 3$). Unpaired and two-sided, *t*-test, *$p = 0.0473$ (GraphPad). **b** ICAM-1 protein expression. *h*SFs were incubated with media alone (Ctr), or with pooled *Pv*EVs or *h*EVs, LPS or TNF-α for 48 h. ICAM-1 expression in *h*SFs at protein level was measured by LSR-Fortessa cytometry. Data show mean of the fold-change relative to control values ± SD (technical replicates, $n = 3$). Unpaired and two-sided, *t*-test, ***$p = 0.0005$, ****$p < 0.0001$ (GraphPad). **c** NF-kB nuclear translocation. *h*SFs were pretreated with media alone or with 5 μM Bay11-7082 for 1 h and incubated with pooled *h*EVs or *Pv*EVs for 30 min. Confocal images (×60 oil) of *h*SFs. Nuclear translocation was determined by nucleus staining (blue) and NF-kB staining (green). Scale bar: 10 μm. Image representative of three independent experiments. **d** ICAM-1 protein expression is blocked by Bay11-7082. Bay11-7082 pretreated (+) or non-treated (−) *h*SFs were incubated with media alone (Ctr) or with pooled *h*EVs or *Pv*EVs for 48 h. ICAM-1 expression in *h*SFs at protein level was measured by LSR-Fortessa cytometry. Data show mean of the fold-change relative to control values ± SD (technical replicates, $n = 3$). Unpaired and two-sided, *t*-test, ****$p < 0.0001$ (GraphPad). Source data are provided as a Source data file.

## Discussion

As intercellular communicators, extracellular vesicles seem to hold precious insights into physiopathology of infectious diseases. Unfortunately, most of this information in malaria has been obtained from indirect associations of circulating microparticles in human infections with disease severity as well as from extrapolations of studies using EVs obtained from in vitro cultures or animal models. Here, to get insights into the physiological role of circulating EVs during human infections, we isolated EVs from patients infected with *P. vivax* during their acute attacks and demonstrated that they specifically interact with human spleen cells facilitating parasite adherence; thus, suggesting a paradigm shift in *P. vivax* biology towards deeper studies of the spleen during infections.

Isolation and purification of circulating EVs in plasma was accomplished by size-exclusion chromatography. This single-standing technology is probing a robust method to isolate plasma-derived EVs as they are largely depleted of abundant plasma proteins and contain several different exosomal markers[25]. Indeed, EVs from vivax patients contained significant higher amounts of CD9, CD63, CD81 and CD71 as compared with circulating EVs from healthy individuals. Of note, the transferrin receptor (CD71) had the highest MFI values which are consistent with the fact that CD71 is one of the most abundant proteins identified in proteomics analysis of reticulocyte-derived EVs[27]. These data thus suggest that the higher concentration of circulating EVs during acute *P. vivax* attacks are likely due to EVs coming from infected CD71[+] reticulocytes. However, we recognize that CD71 is also expressed by other cells, and the existence of unique cell-specific EV markers is still a matter of debate[32,33]. Thus, the reticulocyte-origin of plasma *Pv*EVs circulating during acute vivax attacks remains to be fully demonstrated. Regardless, the high abundance of EVs circulating in patients compared to healthy donors is in agreement with previous observations from natural *Plasmodium* infections[3,34].

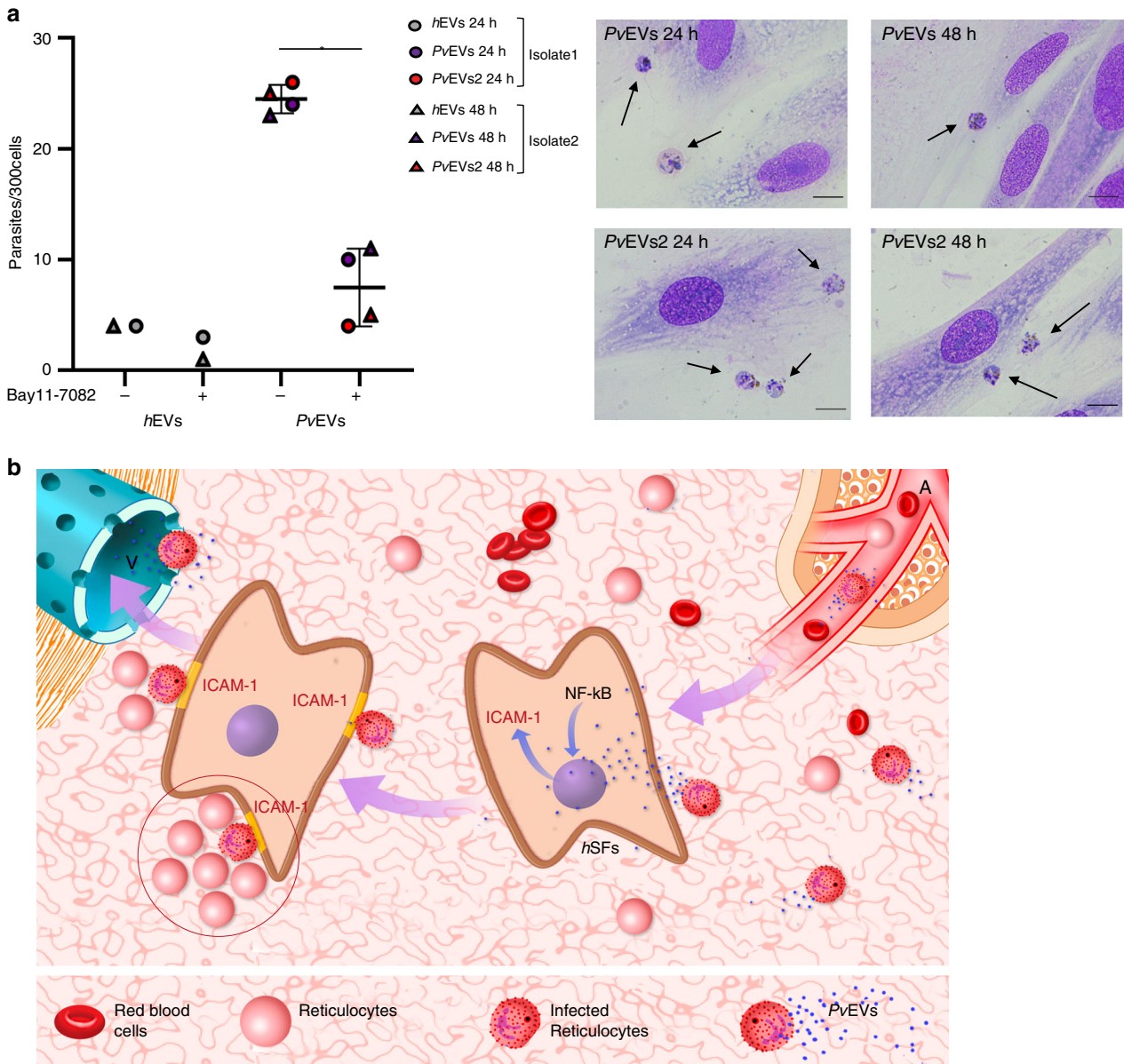

**Fig. 4 Binding of *P. vivax*-infected reticulocytes to *h*SFs is NF-kB dependent. a** Left panel: *h*SFs were cultured in the absence (−) or presence (+) of Bay11-7082 for 1 h and taken up experiments of pooled *h*EVs, *Pv*EVs or *Pv*EVs2 by *h*SFs made at 24 or 48 h, as indicated. Mature parasites from two different isolates (1 or 2) were added to pre-stimulated *h*SFs for 45 min. Bound parasites in 300 cells were counted twice for each condition by two different researchers. Data show mean ± SD ($n = 2$—*h*EVs and $n = 4$—*Pv*EVs). Two-sided, Mann–Whitney test *$p = 0.0286$ (GraphPad). Source data are provided as a Source Data file. Right panel: phase contrast images of Giemsa-staining *h*SFs were captured by optical microscopy (×100/1.25 oil). Arrows show *P. vivax* parasites. Scale bars: 10 μm. Each image represents each independent experiment as indicated. **b** Spleen model of the physiological role of EVs in *P. vivax*. Circulating plasma-derived extracellular vesicles from infected cells (*Pv*EVs) enter the microvasculature of the spleen and are uptake by human spleen fibroblasts (*h*SFs). This event signals NF-kB for translocation to the nucleus and transcription of ICAM-1. After translation of ICAM-1 and expression at the surface of *h*SFs, infected reticulocytes bind to these cells in a niche rich in reticulocytes (circled) where parasites can invade and multiply. A, arteriole; V, venule.

In vivo distribution studies of EVs in mouse models have shown that EVs are preferentially taken up by the liver, spleen and lungs, organs of the mononuclear phagocytic system[35]. Our results are thus in accordance with this preferential uptake of circulating EVs by the spleen and the liver. However, spleen and liver uptake was three-fold higher in animals injected with *Pv*EVs as opposed to animals injected with similar concentration of EVs obtained from healthy individuals or from plasma-derived EVs from cows infected with *Fasciola hepatica*, another parasitic disease. *P. vivax* is one of the few species that has dormant-liver stage forms, called hypnozoites[36] and we had previously

demonstrated adherence of iRBCs to barrier cells of fibroblastic reticular origin in the BALB/c–*P. yoelii* rodent malaria model[19]. Therefore, in addition to the physiological taken up of EVs by the mononuclear phagocyte system of these organs, we speculate that EVs containing *P. vivax*-parasite cargo possibly act as intercellular communicators in the pathophysiology of these organs during infections.

The fact that circulating EVs from malaria infections act as spleen intercellular communicators was initially shown in a rodent malaria model[9,37] and it is confirmed here by the preferential uptake of EVs from infections as opposed to EVs from

normal individuals by human spleen fibroblasts. Specificity of these results was demonstrated by performing competition uptake experiments with EVs from normal individual and infected patients labelled with different lipophilic dyes after extensive washing to remove background. This uptake induced a significant increase in the surface expression of ICAM-1 concomitant with the translocation of the NF-kB transcription factor to the nucleus and this expression was inhibited by Bay11-7082, a specific inhibitor of NF-kB. Further specificity was shown in human endothelial spleen cells where NF-kB translocates to the nucleus under normal conditions and this translocation does not alter expression of ICAM-1. Of note, we developed a semi-automated quantitative assay to measure NF-kB nuclear translocation; thus, facilitating functional analysis of EVs from different sources and this widely used signalling mechanism. Altogether, these results strongly suggest that *Pv*EVs from natural infections are uptake by human spleen fibroblasts, signalling the translocation of the NF-kB transcription factor to the nucleus and upregulation of ICAM-1 surface expression.

Evidence is rapidly accumulating to support the existence of *P. vivax*-parasite populations in the spleen[20,21,24]. Sequestered populations can thus explain the apparent discrepancy of low peripheral blood parasitemia and severe disease. Of note, extra-medullary haemopotopoiesis in the spleen is readily observed during chronic haemolytic anemias[38], and natural *P. vivax* infections in the bone marrow impair erythropoiesis[39,40]. Our data are in agreement with a model where EVs from *P. vivax* infections can enter the microvasculature of the red pulp of the spleen favouring the encounter with and specific uptake by spleen fibroblasts, signalling NF-kB nuclear translocation and increase expression of ICAM-1. These events facilitate the formation of reticulocyte-rich niches where the parasites can cytoadhere and multiply while not circulating in the peripheral blood (Fig. 4b). Together with hypnozoite reservoirs and bone marrow infections[39,41], this hidden parasite population may represent an added challenge for malaria elimination, particularly since cytoadherence to the microvasculature of the spleen will provide a privileged niche for evasion of control measurements[42]. Moreover, such infections may go unrecognized and remain untreated allowing for ongoing transmission of the parasites via mosquito bites; thus, prompting a paradigm shift in *P. vivax* biology towards deeper studies of the spleen during infections and on the physiological contribution of EVs to malaria pathogenesis.

## Methods

**Human plasma samples**. Plasma from *P. vivax* patients used in this study were collected at the tertiary Hospital of the Fundaçao de Medicina Tropical Dr. Heitor Vieira Dourado (FMT-HVD), in the Amazon State, Brazil and in the E.S.E. Hospital San José de Tierralta, Colombia (Supplementary Data 1). These studies were approved, respectively, by the local ethical committees of FMT-HVD in Brazil and Universidad de Córdoba, Monteria in Colombia and written informed consent was obtained from all the participants. Samples from healthy donors were collected at the Hospital Germans Trias i Pujol, Badalona, Spain, after expressed consent from the donors. Ten millilitres of peripheral blood were collected in sodium citrate tubes. Samples were centrifuged at $400 \times g$ for 10 min at RT. Plasma were collected and further centrifuged at $2000 \times g$ for 10 min. Plasma supernatant were recovered, aliquoted in 1 mL fractions and frozen at $-80\,°C$.

**Extracellular vesicles (EVs) isolation**. EVs from plasma samples of healthy donors (*h*EVs), and *P. vivax* patients (*Pv*EVs), as well as from *Fasciola hepatica*-infected cattle from a local slaughterhouse (*Fh*EVs) were isolated by size-exclusion chromatography (SEC)[25]. One-millilitre aliquots of plasma were thawed on ice and centrifuged at $2000 \times g$ for 10 min at 4 °C. Supernatants were loaded on top of 10 mL commercial sepharose 2B columns (iZon Sciences), pre-equilibrated with sterile PBS prepared with LC-MS quality water (Sigma). Fifteen fractions of 500 μL each were immediately collected in 1.5 mL low-protein retention tubes (Eppendorf) with the sterile PBS as the elution buffer and aliquots frozen at $-80\,°C$ until use. EV-enriched SEC fractions from several *P. vivax* patients or healthy donors were pooled to obtain enough amount of EVs for characterization and functional studies.

**EVs quantification**. Protein concentration was measured by BCA Protein Assay Kit (Pierce, Thermo Fisher Scientific). Number and size distribution of particles were determined by Nanoparticle tracking analysis (NTA) in a NanoSight LM10-12 instrument (Malvern Instruments Ltd, Malvern, UK) equipped with a 638 nm laser and CCD camera (model F-033), and data were analysed with NTA software (version 3.2)[25]. Samples were 80 times diluted in PBS to obtain the optimum measurement range of $1 \times 10^8$–$1 \times 10^9$ particles per mL (20–100 particles per frame). Each measurement was performed in technical triplicates.

**Beads-based flow cytometry**. EV pools were identified by beads-based flow cytometry assessing the presence of CD9, CD63, CD81, GAL3, CD5L or/and CD71[43]. The following dilutions of antibodies were used: hybridoma supernatants of CD9 (VJ1/20) 1/7; CD63 (TEA3/10) 1/10; CD81 (5A6) 1/10; commercial GAL3 (Abcam: ab84036) 1/200; CD5L (Abcam: ab45408) 1/200; and CD71 (Abcam: ab84036) 1/1000. Anti-rabbit IgG (Invitrogen: A11008) and anti-mouse IgG (Southern Biotech: 1032-02) secondary antibodies conjugated to FITC were used at 1/1000 and 1/100, respectively. Negative controls included SEC fractions predicted to contain a high concentration of EVs (F7 or F8) and incubated with or without rabbit or mouse IgG isotype, and the respective secondary antibodies. Labelled EV-beads were analysed in a FacsVerse cytometer (BD) and Flow Jo software was used to analyse the data. EV-coupled beads were gated according to FSC-A/SSC-A. Expression of markers was compared using the median fluorescence intensity (MFI) of the beads population (Supplementary Fig. 9). Each measurement was performed in technical triplicates. In total, 10,000 events/sample were analysed.

**Mass spectrometry**. Two-hundred-microlitre aliquots of EV-enriched SEC fractions from samples of 10 *P. vivax* patients and of 10 healthy donors were individually pooled. Pools were processed for EV solubilization and protein digestions[26]. Samples were incubated at 70 °C with RIPA buffer [(50 mM Tris pH 8, 150 mM NaCl, 1 mM EDTA, 0.5% NP-40, 10 mM MgCl₂, 0.5 mM DTT, 1:100 protease inhibitor cocktails (Thermo Scientific)] at an equal ratio (v/v). Samples were sonicated (Bioruptor, Diagenode) for 10 min every 30 s at the highest intensity. Samples were centrifuged at $16,000 \times g$ during 15 min at 4 °C and supernatant was recovered. Proteins were precipitated with cold acetone at a 1/6 ratio (v/v) overnight at $-20\,°C$ and recovered after subsequent centrifugation at $16,000 \times g$ during 15 min at 4 °C. Precipitated proteins were resuspended in 6 M urea, reduced in 10 mM DTT (Sigma) and alkylated with 55 mM of iodoacetamide (Sigma). Samples were brought to a concentration of 2 M urea and digested with a concentration of Lys-C (Wako) corresponding to 10% of the sample (μg) overnight at 37 °C. Samples were further diluted to 1 M urea and digested with a concentration of trypsin (Promega) corresponding to 10% of the sample (μg) for 12 h. Samples were desalted on MicroSpin C18 columns (The Nest Group), evaporated to dryness and dissolved in 0.1% formic acid. One microgram of each sample was analysed by LC-MS/MS using a 90-min gradient in the Orbitrap Fusion Lumos (Thermo Fisher Scientific). As quality control, BSA samples were run between each sample to avoid carryover and to assess instrument performance.

**Protein identification and in silico analysis**. Raw mass spectrometry data files were analysed with the database search algorithm Mascot v2.5.1 (http://www.matrixscience.com) on the Proteome Discoverer software v2.0 (Thermo Fischer Scientific) using a customized protein *P. vivax* database including all sequences from the strains deposited at UniProt and the UniProt human database, downloaded on November 2016. Peptides and proteins were filtered based on (i) minimum peptide length of 7; (ii) maximum false discovery rate (FDR) for peptides and proteins of 1%; (iii) minimum unique peptides per protein of 1; (iv) being classified as master protein. The filtered list of human proteins were compared to the public extracellular vesicle Vesiclepedia plasma protein database[44] (http://www.microvesicles.org/) and intersections were represented using the FunRich tool[45]. Gene ontology enrichment of human proteins exclusively present in *P. vivax* patients was performed with the Database for Annotation, Visualization and Integrated Discovery (David 6.8)[46].

**Western blot analysis**. Prior to western blot analysis, *Pv*EVs obtained from individual patients (Pv1-7 and Pv10) and *h*EVs obtained from individual donors (H1 and H3) were concentrated with Amicon Ultra 0.5 mL 10 K (Millipore) following manufacturer's instructions. Truncated MSP3.1 and PHISTc proteins fused to GST were used as positive controls. Equal amounts of protein (5 μg) were mixed with reducing loading buffer, heated at 95 °C for 5 min and separated on 8% SDS-PAGE gels. Proteins were transferred to nitrocellulose membranes (Amersham) and incubated overnight in blocking buffer (1X PBS, 0.1% Tween-20, 5% milk powder). After three washes with washing buffer (1X PBS, 0.1% Tween-20), blots were incubated for 1 h with primary antibodies: rabbit anti-PvPHIST81[29] at 1:500 dilution, and rabbit anti-PvMSP3[28] serum at 1:500 dilution in a buffer containing 1X PBS, 0.1% Tween-20, 1% milk powder. Antibodies were produced by one of us, MRG, as part of NIH R01 grants (RO1A124710 and RO1AI0555994). After washings, blots were incubated for 1 h with secondary antibodies (IRDye® 680LT Goat anti-Rabbit 925-68021, LICOR Biosciences) at 1:20,000 dilution, and the signal was detected on a LICOR Odyssey Infrared Imaging System using default settings, with the exception of 700-laser intensity, which was set up at 5 for PHISTc

detection and at 3 for MSP3.1 detection. Images were edited using the software ImageJ (NIH).

**EV-labelling for in vivo distribution and cellular uptake.** Lipophilic fluorescence dyes were used to label EVs. For biodistribution experiments, EVs were labelled with CellVue® Burgundy (LICOR). CellVue Burgund (LICOR), is a lipophilic fluorescent dye, which consist of long aliphatic hydrocarbon tails linked to a polar fluorescent chromophore, and integrate into the phospholipid region of cell membranes. For individual cellular uptake experiments, *Pv*EVs and *h*EVs were separately labelled with PKH67 labelling mini kit (Sigma). For competitive uptake experiments, *Pv*EVs and *h*EVs were, respectively, labelled with PKH67 or PKH26 labelling mini kit (Sigma). For STED microscopy, *Pv*EVs were labelled with Abberior STAR 580-DPPE (Abberior GmbH, Göttingen, Germany). Up to 100 µg of EVs in 1 mL of Diluent C were gently mixed for 5 min with 4 µL of dye in 1 mL of Diluent C[37]. In order to remove excess dye, labelling samples were centrifuged for 10 min at $4000 \times g$ on Amicon 100-kDa cut-off filters (Millipore) and either (i) washed five times by centrifugation at $4000 \times g$ for 5 min: twice with 1 mL of PBS and three more times with 100 µL of PBS, (ii) washed one time by centrifugation at $4000 \times g$ for 5 min with 1 mL of PBS, (iii) labelling samples were diluted with 2 ml of EVs-depleted PBS-BSA 1% solution and washed twice by ultracentrifugation at $120,000 \times g$ for 90 min. As a control, PBS was labelled and washed using the same methodologies (Supplementary Fig. 4).

**In vivo distribution of *Pv*EVs, *h*EVs and *Fh*EVs.** For in vivo experiments, 6-week-old C57BL/6 males and a 6-week-old male inmunodeficient nude mouse were used. All animals were housed in cages with enrichment items located in ventilated racks at $21 \pm 1\,°C$ and 50–60% humidity, with 12 h-light/12 h-dark cycles. Food and water were provided ad libitum. Experiments were approved by the ISCIII Ethical Committee and Comunidad Autonoma de Madrid. Three micrograms of Burgundy-labelled-EVs (*Fh*EVs, *h*EVs and *Pv*EVs), quantified by BCA and corresponding, respectively, to $9.08E + 9$, $1.50E + 10$ and $1.38E + 10$ particles as measured by NTA, were used. EVs in a total volume of 100 µL PBS were injected via the retro-orbital venous sinus into C57BL/6 mice ($n = 4$ per group). Burgundy-stained media was also injected to C57BL/6 mice as background control ($n = 4$). In addition, Burgundy-labelled-EVs (*Fh*EVs, *h*EVs and *Pv*EVs) were injected to C57BL/6 mice ($n = 1$) and immunodeficient nude mice ($n = 1$) in the same condition. After 1 h, mice were sacrificed and organs were harvested for the analysis (spleen, lung, heart, bone, brain, gut, pancreas, liver). Harvested organs were imaged in the IVIS-SPECTRUM imaging system (PerkinElmer) with the setting (Level = High, Em = 840 nm, Ex = 745 nm, Epi-illumination, Exposure time = 5 s). To process the images, the same intensity (Min = $5.60E + 6$, Max = $6.4E + 7$) was used in all the experiments. Average radiance was used to quantify the signal of EVs.

**In vivo distribution of *Py*EVs and *m*EVs.** Reticulocyte-derived EVs of *P. yoelii*-infected mice (*Py*EVs) and plasma-derived EVs of healthy mice (*m*EVs) were obtained from BALB/c mice uninfected or infected with *P. yoelii* 17X strain[8,9]. Blood was collected from BALB/c phenylhydrazine-treated mice and of mice infected with *P. yoelii* 17X strain at 20–30% parasitemia in citrate. The blood was thoroughly mixed and centrifuged at $900 \times g$ for 30 min at 4 °C. Plasma was further centrifuged twice at $2000 \times g$ for 10 min at 4 °C and supernatant was kept at −80 °C until use. Cells were washed twice with PBS, diluted up to 50% haematocrit and layered on top of a Percoll-NaCl (GE Healthcare) density gradient as follows: 5 mL of 1.096 g/mL density Percoll solution were placed in 15-mL tubes. Then, 2 mL of 1.058 g/mL density Percoll solution were added. Thereafter, 2 mL of cells at 50% haematocrit were carefully layered on top. Tubes were centrifuged for $250 \times g$ for 30 min at 4 °C. Reticulocytes were collected from the interface of the two Percoll solutions, washed twice with PBS and cultured for 24 h at 1–3% haematocrit at 37 °C in DMEM medium (Sigma) supplemented with 3% EV-depleted FBS (Sigma) and 50 U/mL penicillin–50 µg/mL streptomycin (Gibco). To remove exogenous EVs in the culture medium, the FBS had been previously centrifuged at $100,000 \times g$ for 16 h at 4 °C. After culture, cell viability was assessed with trypan blue (Sigma) staining on a hemocytometer chamber. Reticulocyte cultures with a cell viability >80% were centrifuged at $500 \times g$ for 10 min at RT. Cell-free culture supernatants were further centrifuged at $2000 \times g$ for 10 min at 4 °C and supernatant was used for EV isolation. To purify EVs from either *Py*EVs or *m*EVs, 1 mL of culture supernatant from *P. yoelii*-infected reticulocytes or 1 mL of plasma were processed by SEC following the same procedure described for the isolation of *h*EVs, *Pv*EVs and *Fh*EVs. Ten µg of CellVue® NIR815 (LICOR)-labelled EVs (*Py*EVs and *m*EVs) in a total volume of 100 µL PBS, quantified by BCA and corresponding, respectively, to $2.75E + 7$, $5.46E + 7$ particles as measured by NTA, were injected via the retro-orbital venous sinus to C57BL/6 mice ($n = 3$). After 1 h, mice were sacrificed and organs were harvested for the analysis. The images of each organ were captured by IVIS-SPECTRUM imaging system (PerkinElmer) with the setting (Level = High, Em = 840 nm, Ex = 745 nm, Epi-illumination, Exposure time = 5 s) and the intensity (Min = 5.60E6, Max = 6.4E7).

**Human spleen fibroblasts (*h*SFs).** Spleen samples from transplantation donors were obtained from the Transplant Programme at the Hospital Germans Trias i Pujol. Donation of these organs for use in biomedical research received written consent from family members and was in accordance with the protocol approved by the Ethics Committee for Clinical Research of the Hospital Germans Trias i Pujol. Human spleen fibroblasts (*h*SFs) were obtained from long-term splenocytes cultures[47]. Single-cell suspensions of the spleen were cultured in Dulbecco's modified Eagle's medium (DMEM) (Sigma) supplemented with 10% foetal bovine serum (FBS) (Gibco) and 1% penicillin/streptomycin solution (Gibco) at 37 °C, 5% $CO_2$. After 7 days, cell suspension was removed and only adherent cells were kept in culture until 100% of confluence. Cells were dissociated with a 0.05% trypsin-EDTA solution (Sigma) to seed in new flasks up to 15 times.

For phenotypic characterization, *h*SFs at 70–80% confluency were trypsinized and 2.5 x $10^4$ cells/well resuspended into 500 µL of DMEM supplemented with 10% FBS and seeded on coverslips (Nunc™) in 24-well plates (Merck millipore). Cells were stained with Hoechst® 33342 (Thermo Fisher Scientific), Cell Mask™ (Thermo Fisher Scientific), and CD54-Alexa Fluor® 488 (HCD54, Biolegend) 1/400, overlaid with SlowFade Diamond mounting medium (Thermo Fisher Scientific), sealed and imaged using confocal microscopy (Zeiss LSM 710) with ×40/NA1.30 and ×63/NA1.40 oil-immersion objectives. Further, cells were stained using the following antibodies: CD90-PE/Cy7 [5E10] (Biolegend, Cat#328124) 1/100; CD44-FITC [KM201] (Abcam Cat#ab25340) 1/100; CD54-Alexa Fluor® 488 [HCD54] (Biolegend, Cat#322714) 1/400; CD71-PE [AC102] (Miltenyi Biotec, Cat#130-091-728) 1/400; CD45-PerCP [2D1] (BD Biosciences, Cat#345809) 1/50. Labelled cells were acquired in a LSR-Fortessa cytometer (BD), and Flow Jo software was used to analyse the data. The cell population was selected by FSC-A/SSC-A and cell debris and dead cells were excluded. Afterwards, singlet was selected by FSC-A/FSC-H. A histogram (corresponding fluorescence/count) was used to compare the positivity of markers of interest with unstained cells (Supplementary Fig. 9). In total, 10,000 events/sample were analysed.

**EVs uptake experiments.** *h*SFs at 70–80% confluency were trypsinized and 2.5 x $10^4$ cells/well resuspended into 500 µL of DMEM supplemented with 10% EVs-depleted FBS and seeded on coverslips (Nunc™) in 24-well plates (Merck Milli-pore). 10 µg/mL of PKH67-labelled *h*EVs and *Pv*EVs were individually incubated with cells at 37 °C, 5% $CO_2$ for 30 min, 2 h, and 5 h. After incubation, cells were washed three times with PBS, fixed with 4% paraformaldehyde (Sigma), stained with Hoechst® 33342 (Thermo Fisher Scientific) and Cell Mask™ (Thermo Fisher Scientific), overlaid with SlowFade Diamond mounting medium (Thermo Fisher Scientific), sealed and imaged using confocal microscopy Zeiss LSM 710 with a ×63/NA1.40 oil-immersion objectives. As control of active uptake, EVs were incubated with *h*SFs at 4 °C or with fixed *h*SFs at 37 °C for 2 h. For competition assays, 10 µg/mL of PKH67-labelled *Pv*EVs and PKH26-labelled *h*EVs were incubated together with *h*SFs at 37 °C, 5% $CO_2$ for 2 h and imaged by confocal microscopy. For STED microscopy, 10 µg/mL of Abberior STAR 580-DPPE-labelled *Pv*EVs were incubated with *h*SFs at 37 °C, 5% $CO_2$ for 2 h.

**Confocal microscopy.** Confocal images were acquired on a Zeiss LSM 710 Confocal Module coupled to the Zeiss Axio Observer Z1 microscope with a ×63/NA1.40 oil-immersion objective, equipped with seven lines of lasers (405, 458-488-514, 543, 594 and 633 nm). For processing images, the ZEN black 2012 image acquisition and analysis software was used. The intensity of all the channels was standardized through all the experiments. For quantification assays, images of nine different cells were randomly captured at $1024 \times 1024$ pixel resolution by confocal microscopy and measured by ImageJ version 1.52j (NIH). A fixed threshold was used to remove the low-intensity background signal in all the channels. EVs in each cell were quantified by fluorescence intensity. Cell size (pixel²) was defined according to the cell membrane staining. The intensity (/pixel²) of both EVs was used for the measurement.

**STED microscopy.** Super-resolution analysis of EVs structures was performed using Leica SP8 STED 3X microscope (Mannheim, Germany) equipped with a ×100/1.4NA oil-immersion STED objective. Confocal images of ICAM-1 were acquired using 498 nm excitation line whereas STED and confocal images of labelled EV membranes (Abberior STAR 580-DPPE) were acquired using 587 nm excitation line from the white light laser. Abberior STAR 580 signal was depleted with a donut-shaped 775-nm pulsed STED laser. STED and confocal images were acquired with following parameters: pinhole size: 1 Airy; dwell time: 1.2 µs/pixel, XY pixel size: 20 nm. Acquired images were smoothed, cropped and thresholded using Fiji (ImageJ distribution) software.

**RNA extraction of EVs-stimulated *h*SFs.** *h*SFs at 70–80% confluency were trypsinized and 2.5 x $10^4$ cells/well seeded in 500 µL of DMEM containing 10% EV-depleted FBS in 24-well plates (Merck Millipore) and incubated with *Pv*EVs or *h*EVs (10 or 20 µg/mL) at 37 °C, 5% $CO_2$ for 24 h. *h*SFs without any stimulation were used as control. Each condition was performed in triplicate. Cells were resuspended in 1 mL of Trizol (Invitrogen) and kept at −80 °C until use. Two-hundred microlitres of chloroform (Sigma) was added and vortex for 15 s. After 15 min, samples were centrifuged at $12,000 \times g$ for 15 min at 4 °C. Five hundred microlitres of isopropanol (Sigma) was added to the supernatant and incubated overnight at 4 °C. Samples were centrifuged at $12,000 \times g$ for 10 min at 4 °C, and

washed with 1 mL–75% ethanol (Merck) at $7500 \times g$ for 5 min at 4 °C. The pellet was resuspended in 20 µl of DNase/RNase free water (Invitrogen) and gently vortexed. Two microlitres of sample was used for RNA quality (RIN > 5) and mRNA quantification in an Agilent 2100 Bioanalyzer.

**Quantitative real-time PCR (qRT-PCR)**. Genomic DNA contamination was removed by a DNase digestion step (Thermo Fisher Scientific). Total complementary DNA (cDNA) was synthesized using Superscript IV first strand synthesis system kit (Thermo Fisher Scientific). Quantitative real-time PCR reactions were performed in technical triplicate with the Light Cycler 480 Roche system (Life Science). qRT-PCR reactions were prepared with 5 µL of TaqMan® Fast Advanced Master (Thermo Fisher Scientific), 0.5 µL of each amplification primer, 2.5 µL of nuclease-free water and 2 µL of cDNA templates. PCR programme was initiated with UNG incubation at 50 °C for 2 min and polymerase activation at 95 °C for 20 s, followed by 45 cycles of denaturing at 95 °C for 1 s, and subsequently annealing at 60 °C for 20 s, and 2 µl of nuclease-free water instead of cDNA was used as negative control. Primers used (Thermo Fisher Scientific) are shown below. For the analysis, *guanine nucleotide-binding protein subunit beta-2-lile 1 (GNB2L1)* was used as endogenous control. The expression level ($\Delta Cp$) of each target gene, *alpha-actin-2 (ACTA2)*, *granulocyte-macrophage colony-stimulating factor (GM-CSF)*, *vascular endothelial growth factor A (VEGFa)*, *C-X-C motif chemokine ligand 12 (CXCL12)*, *fibroblast growth factor 8 (FGF8)*, *intercellular adhesion molecule 1 (ICAM-1)*, *interleucine-6 (IL-6)*, *interleucine-10 (IL-10)*, *C-C motif chemokine 2 precursor (CCL2)*, *Toll-like receptors 4, 7, 9 (TLR4, TLR7, TLR9)*, was calculated as $2^{(CtGNB2L1 - Ct\ target\ gene)}$, and then fold-change expression ($\Delta \Delta Cp$) was calculated as the ratio between the expression in EVs-treated cells versus non-stimulated control cells. The name of each gene and the link of each gene used by the Company are shown in Supplementary Table 1.

**Flow cytometry analysis**. *h*SFs at 70–80% confluency were trypsinized and 2.5 x $10^4$ cells/well seeded in 500 µL of DMEM containing 10% EV-depleted FBS in 24-well plates (Merck Millipore) and incubated with *Pv*EVs, *h*EVs (20 µg/mL or 60 µ/mL), LPS (Sigma) or TNFα (Gibco) (100 ng/mL, respectively) at 37 °C, 5% CO$_2$ for 48 h. *h*SFs without any stimulation were used as control. Each condition was performed in triplicate. Cells were detached by accutase (Sigma) and stained using CD54-Alexa Fluor® 488 (HCD54, Biolegend) 1/400. Labelled cells were acquired in a LSR-Fortessa cytometer (BD), and Flow Jo software was used to analyse the data. The population of the cells was selected by FSC-A/SSC-A and cell debris and dead cells were excluded. Afterwards, singlets were selected by FSC-A/FSC-H (Supplementary Fig. 9). Median fluorescence intensity was used for the quantification. In total, 10,000 events/sample were analysed. Fold-change ICAM-1 expression was calculated as the ratio between the expression in stimulated cells versus non-stimulated control cells.

**NF-kB inhibition assay**. *h*SFs at 70–80% confluency were trypsinized and 2.5 x $10^4$ cells/well seeded in 500 µL of DMEM containing 10% EV-depleted FBS on coverslips (Nunc™) in 24-well plates (Merck Millipore). For inhibition assay, cells were treated with 5 µM Bay11-7082 (Sigma) for 1 h. Pretreated and non-pretreated cells were incubated with *Pv*EVs, *h*EVs (20 µg/mL) or medium alone (Ctr) at 37 °C, 5% CO$_2$ for 30 min for nuclear translocation analysis. For flow cytometry analysis, pretreated or not pretreated cells seeded in 24-well plates (Merck Millipore) were incubated with *Pv*EVs, *h*EVs (60 µg/mL), or medium alone (Ctr) at 37 °C, 5% CO$_2$ for 48 h. Each condition was performed in triplicate.

**NF-kB translocation assay**. Treated *h*SFs were fixed by 4% paraformaldehyde for 15 min, permeabilized with 0.1% Triton™ X-100 for 10 min, and blocked with 1% BSA for 1 h. Then, the preparations were incubated with 1:1000 dilution of anti-NF-kB p65 Recombinant Polyclonal Antibody [4-2HCLC] (Thermo Fisher Scientific Cat#710048) for 3 h and 4 µg/mL of Goat anti-Rabbit IgG (H + L), Superclonal™ Recombinant Secondary Antibody, Alexa Fluor 488 (Thermo Fisher Scientific Cat#A27034) for 1 h. Nuclei were stained with Hoechst® 33342 (Thermo Fisher Scientific) and observed by confocal microscopy (Zeiss LSM 710) with 40/NA1.30, or ×63/NA1.40 oil-immersion objectives.

**NF-kB image quantitative assay**. Following staining, images of nuclear and NF-kB signal were acquired using a ZEISS LSM 710 confocal microscope using ×40/NA1.30 oil-immersion objective. Images were acquired with the following parameters: pinhole size—1 Airy unit, 5 × 5 tiled image of combined size 1062 × 1062 µm, XY pixel size—200 nm. Ten different images were randomly captured. Image analysis of NF-kB translocation was assessed using an in-house written macro (see Code availability) for FiJi distribution of ImageJ (https://fiji.sc/). Macro performs following steps: (1) Detection of cell nuclear regions using nuclear staining signal as a guide, (2) Determination of perinuclear region which was defined as 2-µm-thick band surrounding the detected nuclear region, (3) Calculation of integrated density (IntD) value for both nuclear and perinuclear regions of individual cells. The NF-kB translocation ratio for each cell was calculated until reaching 1000 cells using the formula: $NF\text{-}kB\ translocation = ntD_{Nuclear}/IntD_{Perinuclear}$.

**Human spleen endothelial cells (*hSEC*)**. Human spleen endothelial cells (*h*SEC) were purchased from ScienCell (Cat. #5500). *h*SEC were cultured in fibronectin-coated culture vessel using Endothelial Cell Medium (ScienCell) at 37 °C, 5% CO$_2$. Cells were dissociated with 5 times diluted 0.05% trypsin-EDTA solution (Sigma) in DPBS (ScienCell), and seed in new flask up to 15 times. For NF-kB translocation assay, 2.5 x $10^4$ *h*SEC per well were seed in 500 µL of EV-depleted Endothelial Cell Medium on coverslips (Nunc™) in 24-well plates (Merck Millipore), and incubated with *Pv*EVs, *h*EVs (20 µg/mL), or medium alone (Ctr) at 37 °C, 5% CO$_2$ for 30 min. For qRT-PCR, the cells seeded in 24-well plates (Merck Millipore) were incubated with *Pv*EVs, *h*EVs (20 µg/mL), or medium alone (Ctr) at 37 °C, 5% CO$_2$ for 24 h. Each condition was performed in triplicate and qRT-PCR was done in technical triplicates.

**Thawing of cryopreserved *P. vivax* parasites and maturation**. Eight frozen stabilates of Thailand *P. vivax* field isolates were thawed and put in culture[48]. Parasites were thawed in a water bath at 37 °C for 1 min and the volume amount of blood (V) calculated in order to add dropwise 0.1 V of prewarmed 12% NaCl while gently shaking the tube. After 5 min incubation, 10× V of 1.6% prewarmed NaCl were added dropwise and afterwards centrifuged at $500 \times g$ for 5 min. 10× V of 0.9% NaCl were added to the pellet and centrifuged at $500 \times g$ for 5 min. 10× V of McCOY'5A complete medium (McCoy's 5A 12.0 g/L, HEPES 5.96 g/L, NaHCO$_3$ 2.0 g/L, d-glucose 4.0 g/L and 25% of heat inactivate human AB serum) were added and centrifuged at $500 \times g$ for 5 min. Pellets were resuspended in McCOY'5A complete medium to have 5% haematocrit and incubated at 37 °C in a hypoxic environment (5% O$_2$, 5% CO$_2$ and 90% N$_2$). Parasite maturation was checked every 24 h by Giemsa staining.

**Enrichment of *P. vivax* parasites**. After 24 or 48 h maturation, parasite cultures were transferred to 50-mL tubes, centrifuged at $500 \times g$ for 10 min and washed once with 20 mL of RPMI at $800 \times g$ for 10 min. Pellets were resuspended in incomplete RPMI at 20% haematocrit and up to 4 mL gently overlaid onto 4 ml of 60% Percoll and centrifuged at $1200 \times g$ for 20 min without break. Interfaces were collected and washed with iRPMI three times at $500 \times g$ for 10 min. Parasites were counted on a Neubauer chamber and parasite stage was determined by Giemsa stained smear. Of note, after maturation and enrichment, only two out of the eight isolates had sufficient amounts of mature parasites to carry-on the experiments: isolate 1 after 24 h and isolate 2 after 48 h.

**P. vivax-human spleen fibroblasts binding assays**. *h*SFs at 70–80% confluency were trypsinized and 2.5 x $10^4$ cells/well seeded on coverslips (Nunc™) in 24-well plates (Merck millipore) and incubated for 24 or 48 h with 50 µL of *Pv*EVs, with 50 µL of *Pv*EVs2 [another SEC pool obtained from three *P. vivax* patients from Tierralta, Colombia (Supplementary Data 1)], or with 50 µL of *h*EVs at 37 °C, 5% CO$_2$ in DMEM supplemented with 10% EVs-depleted FBS. For inhibition assays, cells were treated for 1 h with 5 µM Bay11-7082 (Sigma) prior to EVs incubation. After EVs incubations, cells were washed with binding medium (RPMI pH 6.8 supplemented with 10% AB serum), and $0.42 \times 10^5$ (isolate 1/ 24 h maturation) and $1.4 \times 10^5$ (isolate 2/48 h maturation) iRBCs were added to each condition in two independent experiments. Parasites were incubated at 37 °C, 5% CO$_2$ for 45 min, and unbound cells were washed three times gently with prewarmed binding medium. Samples were fixed with methanol for 1 min and stained with Giemsa. The number of parasites bound to cells were count twice independently by two different researchers (HT and WR) in 300 cells of each preparation using optical microscopy (Eclipse Ci-L, Nikon) with ×20/0.40 and ×100/1.25 oil objectives.

**Statistical analysis**. All data were analysed using GraphPad Prism software (Version 8.3.0) and are presented as mean ± standard deviation. Figures 1e, 3a, b, d, and Supplementary Fig. 7b was technical triplicate using the pooled EVs samples. As the source of variability presented in experiments in Figs. 1e, 3a, b, d, and Supplementary Figs. 6d, 7b was not biological, demographic or temporal but inherent technical error, *p*-values were assessed using unpaired and two-sided *t*-test assuming normal distributions. In fact, our data did not show signs of kurtosis or presence of outliers. Supplementary Fig. 6d was single analysis of each cell ($n = 1000$) using pooled EVs samples. As the distributions showed normality, *p*-values were assessed using unpaired and two-sided *t*-test. Figure 1a was single analysis of individual EVs sample ($n = 10$), Figure 2b, c was single analysis of each cell ($n = 9$) using pooled samples, Fig. 4a used two different batch of pooled EVs samples for two different *P. vivax* isolates and was count twice independently by two different researchers. Supplementary Fig. 1a was single analysis of individual EVs sample ($n = 10$). The non-parametric, unpaired and two-sided Mann–Whitney test was used to calculate *p*-values in experiments represented in Figs. 1a, 2b, c, 4a, and Supplementary Fig. 1a. Differences were considered significant when the *p*-value was < 0.05. *$p < 0.05$, **$p < 0.01$, ***$p < 0.001$, ****$p < 0.0001$.

**Reporting summary**. Further information on research design is available in the Nature Research Reporting Summary linked to this article.

## Data availability

The mass spectrometry proteomics data have been deposited to the ProteomeXchange Consortium via the PRIDE[49] partner repository with the dataset identifier PXD018337. The source data underlying Fig. 1a, c, e, 2a–c, 3a, b, d, 4a and Supplementary Figs. 1a, 2b, 7b are provided as a Source Data file. All the other data supporting the findings of this study are available in the main text or in supplementary materials.

## Code availability

The ImageJ macro used for NF-kB translocation evaluation was deposited in Zenodo open-access repository with code 3555213 under Creative Commons Attribution 4.0 International license.

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

## Acknowledgements

To all patients and healthy donors that participated of these studies, to Francisco Sánchez-Madrid and Maria Yañez-Mo, UAM, Madrid for the kind gift of CD63, CD9 and CD81 hybridomas, to Marco Fernandez at the cytometry Unit, IGTP for helpful discussions on cytometry, to Victor Urrea at IrsiCaixa for helpful discussions on statistical analysis, to Aleix Elizalde-Torrent for helping with PyREX isolations and to Marc Nicolau for technical assistance. We also thank the Advanced Light Microscopy Unit at

the Centre for Genomic Regulation (CRG, Barcelona, Spain) for access to the Leica STED microscope. H.T. (2017FI_B1_00202) and M.D.V. (2017FI_B2_00029) are predoctoral fellows supported by Secretaria d'Universitats i Recerca del Departament d'Economia iCreixement, Generalitat de Catalunya. M.G.L. is a postdoctoral fellow supported by the Plan Estratégico (PERIS) of the Generalitat de Catalunya. I.A.H. is a predoctoral fellow supported by the Ministerio de Economia y Competitividad (FPI BES-2017081657). J.C. is supported by European Union's Horizon 2020 research and innovation programme under the Marie Skłodowska-Curie grant agreement No. 793830. MSP3 and PHIST antibodies were generated with funding from NIH to M.R.G. (RO1A124710 and RO1AI0555994). The CRG/UPF Proteomics Unit is part of the Spanish Infrastructure for Omics Technologies (ICTS OmicsTech) and it is a member of the ProteoRed PRB3 consortium which is supported by grant PT17/0019 of the PE I+D+i 2013–2016 from the Instituto de Salud Carlos III (ISCIII) and ERDF. We acknowledge support from the Spanish Ministry of Science, Innovation and Universities, "Centro de Excelencia Severo Ochoa 2013–2017", SEV-2012-0208, and "Secretaria d'Universitats i Recerca del Departament d'Economia i Coneixement de la Generalitat de Catalunya" (2017SGR595). This research is part of ISGlobal's Programme on the Molecular Mechanisms of Malaria which is partially supported by the Fundación Ramón Areces. Work in the laboratory of Carmen Fernandez-Becerra and Hernando A del Portillo is funded by the Ministerio Español de Economía y Competitividad (SAF2016-80655-R). ISGlobal and IGTP are members of the CERCA Programme, Generalitat de Catalunya. This work received specific support from the Fundación Ramón Areces, 2014, "Investigación en Ciencias de la Vida y de la Materia", Project "Exosomas: Nuevos comunicadores intercelulares y su aplicabilidad como agentes terapéuticos en enfermedades parasitarias desatendidas".

## Author contributions

H.T., M.D.V., J.S.B., W.R., B.B., S.G.S., A.G., I.A.H., C.C.C., J.C. and C.F.B. performed experiments. H.T., M.D.V., S.G.S., M.G.L., E.B., E.S., I.A., J.C., M.C., P.A., A.M., H.P., C.F.B. and H.D.P. suggested experiments and analysed data. A.A., M.B., G.M., W.M., J.C.F., M.F.Y., R.L., A.M., M.R.G., M.V.G.L. and J.P.S. contributed materials. H.T., M.D.V., C.F.B. and H.D.P. drafted the paper. M.R.G. and J.M.P. provided critical feedback and manuscript revisions, and all authors reviewed the paper and consented to its publication. C.F.B. and H.D.P. conceived this study.

## Competing interests

The authors declare no competing interests.

## Additional information

Haruka Toda[1], Miriam Diaz-Varela[1], Joan Segui-Barber[1], Wanlapa Roobsoong[2], Barbara Baro[3,20], Susana Garcia-Silva[4], Alicia Galiano[5], Melisa Gualdrón-López[6], Anne C. G. Almeida[3,7], Marcelo A. M. Brito[3,7], Gisely Cardoso de Melo[3,7], Iris Aparici-Herraiz[1], Carlos Castro-Cavadía[8], Wuelton Marcelo Monteiro[3,7], Eva Borràs[9,10], Eduard Sabidó[9,10], Igor C. Almeida[11], Jakub Chojnacki[12], Javier Martinez-Picado[6,12,13,14], Maria Calvo[15], Pilar Armengol[6], Jaime Carmona-Fonseca[8], Maria Fernanda Yasnot[16], Ricardo Lauzurica[17], Antonio Marcilla[5], Hector Peinado[4], Mary R. Galinski[18], Marcus V. G. Lacerda[3,19], Jetsumon Sattabongkot[2], Carmen Fernandez-Becerra[1,6✉] & Hernando A. del Portillo[1,6,14✉]

[1]ISGlobal, Hospital Clínic - Universitat de Barcelona, Barcelona 08036, Spain. [2]Mahidol Vivax Research Unit, Faculty of Tropical Medicine, Mahidol University, Bangkok 10400, Thailand. [3]Fundaçao de Medicina Tropical Dr. Heitor Vieira Dourado (FMT-HVD), Manaus, Amazonas 69040-000, Brazil. [4]Microenvironment and Metastasis Laboratory, Department of Molecular Oncology, Spanish National Cancer Research Center (CNIO), Madrid 28029, Spain. [5]Àrea de Parasitologia, Departament de Farmàcia i Tecnologia Farmacèutica i Parasitologia, Universitat de València, Burjassot, Valencia 46100, Spain. [6]Germans Trias i Pujol Health Science Research Institute (IGTP), Badalona 08916, Spain. [7]Universidade do Estado do Amazonas (UEA), Manaus, Amazonas 69020-070, Brazil. [8]Grupo de Salud y Comunidad Cesar Uribe Piedrahíta, Universidad de Antioquia, Medellín, Colombia. [9]Proteomics Unit, Centre de Regulació Genòmica (CRG), Barcelona Institute of Science and Technology (BIST), Barcelona 08003, Spain. [10]Universitat Pompeu Fabra (UPF), Barcelona 08002, Spain. [11]Border Biomedical Research Center, Department of Biological Sciences, College of Science, University of Texas El Paso, El Paso, TX 79902, USA. [12]AIDS Research Institute IrsiCaixa, Badalona 08916, Spain. [13]University of Vic-Central University of Catalonia, Vic 08500, Spain. [14]Institució Catalana de Recerca i Estudis Avançats (ICREA), Barcelona 08010, Spain. [15]Unitat de Microscopia Òptica Avançada, Facultat de Medicina, Centres Científics i Tecnològics, Universitat de Barcelona, Barcelona 08028, Spain. [16]Grupo de Investigaciones Microbiológicas y Biomédicas de Córdoba-GIMBIC, Universidad de Córdoba, Monteria 230001, Colombia. [17]Nephrology Service, Germans Trias i Pujol University Hospital, Badalona 08916, Spain. [18]Emory Vaccine Center, Yerkes National Primate Research Center, School of Medicine, Division of Infectious Diseases, Emory University, Atlanta, GA 30329, USA. [19]Instituto Leônidas & Maria Deane (ILMD), Fiocruz, Manaus, Amazonas 69057-070, Brazil. [20]Present address: ISGlobal, Hospital Clínic - Universitat de Barcelona, Barcelona, Spain. ✉email: carmen.fernandez@isglobal.org; hernandoa.delportillo@isglobal.org

