## [Peer Review File · Nature Communications]

Reviewers' Comments:

Reviewer #1:

Remarks to the Author:

The manuscript submitted by Toda et al focuses on an emerging field in infectious diseases nowadays, investigating the role of extracellular vesicles (EVs), in this case in malaria pathogenesis. This study offers the observation that *Plasmodium vivax* patients have high levels of secreted EVs (PvEVs) in circulation. The authors found parasitic proteins in the EV protein cargo and that the EV distribution (in an animal model) exhibits spleen tropism. They also suggest a possible influence of the EVs on *P. vivax* virulence, by potentially affecting the adhesion of RBCs to spleen fibroblasts and creating a niche for parasite replication and invasion.

Let me start by noting that this work is not only important for the field of parasitology in general but is also novel in two respects:

1. This is the first comprehensive EV study that concentrates on one of the most dangerous human malaria parasite species, *Plasmodium vivax*. Of importance, this species cannot be grown in laboratory conditions, meaning that any basic research work with *P. vivax*-derived samples is extremely challenging and rare, and at the same time is incredibly important.
2. Here, the authors used EV samples derived from malaria patients from endemic areas. To the best of my knowledge, this is one of the first, if not THE first, study to find the physiological "contribution" of EVs to malaria pathogenesis. Again, very ground-breaking work.

For these two main reasons, I believe the manuscript potentially merits publication in *Nature Communications*.

The manuscript is well explained and well written, and the experiments are mostly well planned and focused, which overall reflects the hard work of the research team. Nevertheless, I have some concerns regarding some of the methodological and analytical aspects of the research, and these should be addressed by the authors. Also, additional experiments should be done to further validate the suggested mechanism.

General comments:

- Generally, the statistical analyses throughout the entire work need to be improved and further clarified. The complexity involved in working with pools of infected and control patients requires statistical tests that should be explained in the body of the manuscript, as well as in the Materials and Methods section. In addition, some statistical results are incomplete or not shown in some of the experiments, which are pointed out for each figure in the minor comments section below.
- The use of immunocompetent mice to analyze the biodistribution of PvEVs and hEVs could have influenced the normal physiological distribution of the EVs due to their human origin. Thus, immunodeficient mice strains would provide a better representation of the EV uptake in different organs. The authors should clarify this point.

Minor comments:

- Demographic and clinical information of all the patients is needed. This should include a table presenting the patients' age, gender, clinical signs and parasitemia levels and whether the patients had co-occurring diseases.

Figure 1

- How many repeats were done for the experiments using clinical samples for this figure? Please clarify for each panel. For instance, how many healthy samples were tested for the expression of the proteins in Figure 1d? Are there any variations?
- The EV concentration shown in Fig. 1A seems to be out of the range of the NanoSight (NTA) measurement. Thus, EV concentrations should be diluted in order to obtain a more reliable result and their effect needs to be re-examined to validate the data.
- The error bars in Fig. 1B are missing. Please provide the results of the statistical analyses.
- The Western blot panel shown in Fig. 1D using MSP3.1 and PHIST is missing positive and negative controls. The authors present results derived only from one hEV sample; more patient samples need to be analyzed by WB (minimum of three clinical samples).
- In Line 146, the authors should clarify how were the EVs for the biodistribution experiment quantified? This is not mentioned also in the Methods section.

Figure 2

- EVs were stained with CellVue®. Please elaborate more on the staining specificity of this dye.
- The authors indicate that three micrograms of EVs were used for the biodistribution assay. This form of quantification is problematic and not according to the standard acceptable method. The number of EVs (by NTA) that were administered to the mice should be indicated as well.
- The use of labelled PBS as a background control in the biodistribution experiment may not be the best choice (Fig. 2A). Instead of using PBS as a control, the authors should use stained media or EV-depleted serum to check the background fluorescence in the organs.
- Fig. 2B shows a competition assay of a mixed population of PvEVs vs. hEVs. However, the uptake level of PvEVs and hEVs separately should also be analyzed, as this would demonstrate the contribution of each EV population in the hSF uptake, and then, the magnitude of the increase/decrease after they compete.
- The authors show the in-vivo distribution of the EVs to the spleen and liver. Yet from this point onwards, the authors do not relate to the liver uptake or the role of the vesicles. It would be especially interesting to see what pathways the EVs activate in liver cells.

Figure 3

- Fig. 3A shows the expression of several inflammatory markers detected by RT-PCR in activated hSF. A longer observation period (of more than 24 hours), as is the custom in the field, is needed in order to measure the fold change in the expression of these markers, especially CXCL12 and CCL2.
- Also in Fig. 3A, the involvement of ICAM-1 in the response of activated hSF should be demonstrated also at the protein level. siRNA against ICAM-1, ICAM-1 knockout strains or protein shaving are potential experiments that can demonstrate that the expressed ICAM-1 is also translated into proteins.
- In the legend of Fig. 3B, it is indicated that ICAM-1 expression at the protein level was measured by LSR-Fortessa cytometry, but this assay does not appear in the Methods section. Please explain the procedure and clarify whether it was performed in other experiments.
- In order to confirm the translocation of NF- κ B to the nucleus, as a complementary approach, I strongly suggest that biochemical nuclear fractionation be performed.

Figure 4

- To show that the adhesion increases due to activation of NF- κ B in Figure 4a, it would be nice if the authors use immunofluorescence together with the localization of NF- κ B.
- The same experiments for checking ICAM-1 at the protein level are suggested to demonstrate that ICAM-1 indeed mediates the binding of iRBC.
- In order to prove that the increased adhesion is due to enhanced NF- κ B-dependent ICAM-1 expression, the experiment in 4A should be repeated in the presence of ICAM-1 and NF- κ B inhibitors

(e.g., Selleck biochem A-205804 and Bay 11-7082).

Methods

- Line 397: The IVIS-SPECTRUM system was used to monitor the biodistribution of labelled EVs. Please elaborate on the conditions in which this experiment was performed.
- The protocol explained in Lines 379-384 is not completely clear. I would suggest rephrasing the procedure so as to improve the understanding.
- In Line 115, delete the word 'extracellular' from the text "proteins belonging to extracellular exosomes".
- In lines 432 and 433, controls of EV active uptake are mentioned. Please show the results of these controls within the manuscript.
- Line 471: Please change the word "TagMan" to "TaqMan".
- Line 523: Please change the word "two-tails" to "two-tailed".

Missing References:

- The authors should include the following key references:
 - o Demarta-Gatsi C et al, (2019). 'Histamine releasing factor and elongation factor 1 alpha secreted via malaria parasites extracellular vesicles promote immune evasion by inhibiting specific T cell responses'
 - o Mantel et al., (2016). 'Infected erythrocyte-derived extracellular vesicles alter vascular function via regulatory Ago2-miRNA complexes in malaria'.
 - o Sisquella et al. (2017). 'Malaria parasite DNA-harboring vesicles activate cytosolic immune sensors'.

Reviewer #2:

Remarks to the Author:

Toda et al have studied the interaction between plasma extracellular vesicle and spleen fibroblasts in relation to Plasmodium vivax cytoadhesion and sequestration. This is an interesting article which proposes an interesting hypothesis on the biology and pathogenesis of P. vivax infection. However, although most of the experiments are properly designed and performed, they lack adequate statistical analyses. In addition, there is a certain bias in the analysis of the data and to support the authors' conclusions, additional data are required.

Specific comments

1. The authors used the term "concealed". This means that the parasite is hiding, hiding from what? If the parasite cytoadheres to fibroblast (as shown here) and to endothelial cells (as demonstrated by the same group previously, Carvalho et al, J Infect Dis, 2010, 202, 638), why not use the term cytoadherence and sequestration as proposed by the same authors in a previous articles (Lopez et al, J Infect Dis, 2014, 209, 1403). I propose that they should avoid new terminology unless they provide a rationale for it.
2. The major limitation of this article is its statistical analyses.
3. The authors did not show that their data show the normal distribution and thus cannot use t-test or ANOVA test. The authors should perform Kruskal-Wallis test or Mann-Whitney tests when displaying 4 groups in their histogram plots or two in the main graph.

4. The claim that there is tropism of the EV to the spleen is based on the experiments described in figure 2. This system is artificial and is not conclusive. In any heterologous system, the liver and spleen are the primary sites of filtration. At least a control with EV from infected mice should be used. This system should be validated before any conclusions can be drawn.

5. The authors focus all the remaining on the effect in the EV on human spleen fibroblast. This is a strongly biased approach. They do not rule out the cytoadherence to endothelial or to monocytes/macrophages, or phagocytosis by monocyte/macrophages (see Chamberlain et al, 2019 Stem Cells, 37, 652) may be more relevant for sequestration of the parasite in the spleen. They should perform experiments using endothelial cells and monocytes/macrophages.

6. The claim, line 253, that there is no endothelium in the bone marrow and in the spleen is inaccurate. For the bone marrow: Tavassoli, Prog Clin Biol Res, 1981; Raffi et al., 1995, 86,9, 3353; for the spleen, see: Qiu et al, Blood Advances, 2,1130.

Reviewer #1 (Remarks to the Author):

The manuscript submitted by Toda et al focuses on an emerging field in infectious diseases nowadays, investigating the role of extracellular vesicles (EVs), in this case in malaria pathogenesis. This study offers the observation that *Plasmodium vivax* patients have high levels of secreted EVs (PvEVs) in circulation. The authors found parasitic proteins in the EV protein cargo and that the EV distribution (in an animal model) exhibits spleen tropism. They also suggest a possible influence of the EVs on *P. vivax* virulence, by potentially affecting the adhesion of RBCs to spleen fibroblasts and creating a niche for parasite replication and invasion.

Let me start by noting that this work is not only important for the field of parasitology in general but is also novel in two respects:

1. This is the first comprehensive EV study that concentrates on one of the most dangerous human malaria parasite species, *Plasmodium vivax*. Of importance, this species cannot be grown in laboratory conditions, meaning that any basic research work with *P. vivax*-derived samples is extremely challenging and rare, and at the same time is incredibly important.

2. Here, the authors used EV samples derived from malaria patients from endemic areas. To the best of my knowledge, this is one of the first, if not THE first, study to find the physiological “contribution” of EVs to malaria pathogenesis. Again, very ground-breaking work.

For these two main reasons, I believe the manuscript potentially merits publication in Nature Communications.

We are sincerely grateful to this reviewer for these encouraging comments and for the overall positive reception of this manuscript, and its acknowledged relevance with regards to *P. vivax* malaria.

The manuscript is well explained and well written, and the experiments are mostly well planned and focused, which overall reflects the hard work of the research team. Nevertheless, I have some concerns regarding some of the methodological and analytical aspects of the research, and these should be addressed by the authors. Also, additional experiments should be done to further validate the suggested mechanism.

General comments:

- Generally, the statistical analyses throughout the entire work need to be improved and further clarified. **The complexity involved in working with pools of infected and control patients requires statistical tests that should be explained in the body of the manuscript, as well as in the Materials and Methods section.** In addition, some statistical results are incomplete or not shown

in some of the experiments, which are pointed out for each figure in the minor comments section below.

We appreciate this concern, also shared by the second reviewer. Accordingly, we have reviewed all statistics and have made the following changes:

1. We are aware of the complexity of working with pools of biological samples from different individual subjects ¹. However, the amount of peripheral blood that ethically can be withdrawn from patients during acute attacks made unfeasible to perform all the experiments with EVs from individual patients. Therefore, it was necessary to make a pool of all samples. To demonstrate the complexity of the samples used to make these pools, we have replaced Fig. 1a with a figure that shows the protein concentration from the individual samples. We have also added the following text into the revised version:

*"We used size-exclusion chromatography (SEC) to isolate EVs as this single technology purifies EVs and removes abundant soluble plasma proteins. Moreover, previous studies consistently showed that SEC fractions 7,8, and 9 contained the plasma EVs-enriched fractions. Thus, we isolated plasma-derived EVs from ten P. vivax patients (Table S1) and from 10 healthy donors and obtained individual pools of these SEC fractions from each subject. We measured their protein content and showed a large variability among the different samples (Fig. 1a); yet, a significant difference between samples from infected as opposed to healthy individuals was observed (Mann-Whitney test, *p=0,02).*

2. We have now used the Mann-Whitney test (i) in the results of variability of the individual samples (Fig. 1a), (ii) in the results of the different uptaken assays (Fig. 2b-c), (iii) in the binding/inhibition assays (Fig. 4a) and (iv) in the difference in number of human proteins (Fig. S1).

3. We applied the Kruskal-Kallis test followed by Dunn's multicomparison test to analyse the data on in vivo distribution. As using this statistical test we only reached significant differences between PvEVs and Controls, we decided to remove statistical values from this Figure. Yet, we showed the individual values of each experiment (Fig. 2a).

4. We used unpaired two-tailed *t*-test in the experiments shown in Figures 1e, 3a, 3b, 3d, S6d and S7b as this same test has been used to obtain p-values in similar experiments with n=3 in *Nature Communications* [see for examples: (Abdalle et al., 10,3778 Figs1-2 (2019); Zeng, et al., 9, 5395 Fig 1 (2018); Sisquella et al., 8, 1985 Fig 7 (2017)].

5. We have now shown the distribution of the different experiments as dots in all figures. Differences were considered significant when the *p*-value was <0.05. **p* <0.05, ***p* <0.01, ****p* <0.001, **** *p* <0.0001.

- The use of immunocompetent mice to analyze the biodistribution of PvEVs and hEVs could have influenced the normal physiological distribution of the EVs due to their human origin. Thus, immunodeficient mice strains would provide a better

representation of the EV uptake in different organs. The authors should clarify this point.

There are good examples showing *in vivo* distribution of EVs in immunocompetent mice². Yet, we agree with this reviewer that this might represent a confounding factor. Thus, we performed similar experiments in immunosuppressed mice and showed a similar distribution (Fig. S2). Moreover, we have also demonstrated a similar tropism in mice injected with reticulocyte-derived EVs obtained from BALB/c mice infected with the reticulocyte-prone *P. yoelii* 17N strain (Fig. S2). We have also added the following text in the new manuscript:

“As immunocompetent mice might alter the in vivo distribution of EVs from human and animal origin, we performed similar experiments in immunosuppressed animals and observed the same trend in distribution (Fig. S2). In addition, we also observed similar tropisms using reticulocyte-derived exosomes from BALB/c mice infected with the reticulocyte-prone P. yoelii strain (Fig. S2)”.

Minor comments:

- Demographic and clinical information of all the patients is needed. This should include a table presenting the patients' age, gender, clinical signs and parasitemia levels and whether the patients had co-occurring diseases.

Supplementary Table S1 contains now this information. Notably, patients who are diagnosed with *P. vivax* malaria infections in Brazil and Colombia are immediately treated with antimalarial drugs. Accordingly, no other diagnostic test is performed at the moment of the acute attack. However, patients are programmed to come back for a clinical visit within 28 days after antimalarial treatment. If during this first visit, they show no other clinical symptoms, patients are declared free of other diseases.

The new binding/inhibition experiments required a new pool from other patients as the original pool was not sufficient to obtain 4 replicates required for the new statistical analysis. Information of these patients is also shown in Table S1.

Figure 1

- How many repeats were done for the experiments using clinical samples for this figure? Please clarify for each panel. For instance, how many healthy samples were tested for the expression of the proteins in Figure 1d? Are there any variations?

Figure 1 has now been replaced: Fig 1a represent individual BCA-protein concentrations from pooled SEC fractions (7,8,9) of ten individual patients and ten healthy donors. Fig. 1b shows parasite proteins associated with circulating EVs of ten individual patients. Fig. 1c shows the validation of the presence of these proteins associated with EVs by western blot. Fig. 1d shows the NTA concentration of the pool of tens samples from the individual patients and ten healthy donors using the correct measurement range of the NTA machine (20-100 particles per frame). Fig. 1e shows the results of the bead-based flow cytometer assay with different exosomal markers now performed in triplicates. Error bars and p-values as determined by unpaired *t*-test are now shown.

- The EV concentration shown in Fig. 1A seems to be out of the range of the NanoSight (NTA) measurement. Thus, EV concentrations should be diluted in order to obtain a more reliable result and their effect needs to be re-examined to validate the data.

We apologise for this oversight from our end. We have now repeated all measurements using the correct measurement range of the NTA machine (20-100 particles per frame) and made a new figure including the number of particles and their corresponding protein concentration (Fig. 1d).

- The error bars in Fig. 1B are missing. Please provide the results of the statistical analyses.

Error bars and statistical analysis are now shown in the figure (Fig. 1e).

- The Western blot panel shown in Fig. 1D using MSP3.1 and PHIST is missing positive and negative controls. The authors present results derived only from one hEV sample; more patient samples need to be analyzed by WB (minimum of three clinical samples).

We have performed a new western blot including individual serum from human healthy volunteers (see below). However, to accommodate the results on the NTA and bead-based flow cytometer assays in Figure 1, we kept the original figure including positive controls for the MSP3 and PHIST proteins.

- In Line 146, the authors should clarify how were the EVs for the biodistribution experiment quantified? This is not mentioned also in the Methods section.

This has now been clarified in the text: *“Briefly, 3 μg of Burgundy-labelled-EVs (FhEVs, hEVs, and PvEVs), quantified by BCA and corresponding, respectively, to 9.08E+9, 1.50E+10, 1.38E+10 particles as measured by NTA in a total volume of 100uL PBS”.*

Figure 2

- EVs were stained with CellVue®. Please elaborate more on the staining specificity of this dye.

We have now added the following sentence to elaborate more on this dye: *“CellVue Burgund is a lipophilic fluorescent dye, which consist of long aliphatic hydrocarbon tails linked to a polar fluorescent chromophore, and integrate into the phospholipid region of cell membranes”.*

- The authors indicate that three micrograms of EVs were used for the biodistribution assay. This form of quantification is problematic and not according to the standard acceptable method. The number of EVs (by NTA) that were administered to the mice should be indicated as well.

To the best of our knowledge, there is presently no consensus in the EV-research field on a unique quantitation method of these vesicles. All of the techniques have pros and cons. For instance, NTA counts particles and not only EVs. Yet, as requested by this reviewer, the number of EVs injected into mice, as determined by NTA particles/mL, are now indicated in the legend of Figure 2 and this information was also added in Methods.

- The use of labelled PBS as a background control in the biodistribution experiment may not be the best choice (Fig. 2A). Instead of using PBS as a control, the authors should use stained media or EV-depleted serum to check the background fluorescence in the organs.

We think there is no disagreement with regard to this comment. Labelled-PBS in these experiments means stained medium.

- Fig. 2B shows a competition assay of a mixed population of PvEVs vs. hEVs. However, the uptake level of PvEVs and hEVs separately should also be analyzed, as this would demonstrate the contribution of each EV population in the hSF uptake, and then, the magnitude of the increase/decrease after they compete.

We thank this reviewer for suggesting this experiment. The new data show that EVs from patients are taken up significantly higher in individually uptake experiments (Mann-Whitney $***p < 0.001$) as compared to competition uptake experiments (Mann-Whitney $*p < 0.05$). Both results are now shown in Fig. 2 and the following text add to the manuscript:

*“Next, we labelled individually PvEVs and hEVs with PKH67, a lipophilic membrane fluorescence dye. Worth of mentioning, only after washing five times through Amicon 100-kDa cut-off filters, unbound dye was completely removed avoiding this confounding factor (Fig. S4). Both preparations were incubated with hSFs for 2h, the peak of active uptake (Fig. S5) and images were randomly captured and quantified. Results demonstrated that significantly more PvEVs were taken up by hSFs than hEVs (Mann-Whitney test, $***p < 0.001$) (Fig. 2b). As EVs from non-infected cells and EVs from *P. vivax*-infected reticulocytes circulate together during natural infections, we performed competition taken up experiments by labelling hEVs with PKH26 (red) and PvEVs with PKH67 (green). Results demonstrated that under these competing situations, PvEVs were still preferentially taken up by hSFs than hEVs albeit at lower levels as judged by the significance observed (Mann-Whitney test, $*p < 0.05$) (Fig. 2c)”.*

- The authors show the in-vivo distribution of the EVs to the spleen and liver. Yet from this point onwards, the authors do not relate to the liver uptake or the role of the vesicles. It would be especially interesting to see what pathways the EVs activate in liver cells.

We certainly agree with this comment. Yet, we are sure that this reviewer will understand that this interest is beyond the scope of this article. Presently, in

collaboration with other international groups with access to insectaries, sporozoites and human liver platforms, we are starting to look into funding to pursue studies on signaling/pathways elicited by EVs from *P. vivax* infections in the liver.

Figure 3

- Fig. 3A shows the expression of several inflammatory markers detected by RT-PCR in activated hSF. A longer observation period (of more than 24 hours), as is the custom in the field, is needed in order to measure the fold change in the expression of these markers, especially CXCL12 and CCL2.

We based this analysis on the research paper published by Sisquella et al., (Nat Comm 2017, 8(1):1985 Fig 6) who made a time-course (0.5h, 3h, 6h, 12h and 24h) transcriptional analysis of chemokine genes, among others, upon up-taken of EVs. Based on this time-course, we also performed a time-course study (5h and 24h) of the ICAM-1, CXCL12 and CCL2 genes (see figure below). Unlike ICAM-1 expression, we did not observe any significant increase in the expression of the latter two genes after 24h. Moreover, the new statistical analysis requested by reviewer 2 did not show statistical significance in expression of CXCL12 and CCL2 genes. Last, our analysis of ICAM-1 expression at the surface of hSFs cells demonstrated that by 48h this molecule was already expressed (Fig. 3b). Thus, we decided to use 24h to analyze transcription of all other genes. We hope that this reviewer understands that due to the limited amount of EVs for other experiments requested, we did not repeat these experiments with longer incubation times.

- Also in Fig. 3A, the involvement of ICAM-1 in the response of activated hSF should be demonstrated also at the protein level. siRNA against ICAM-1, ICAM-1 knockout strains or protein shaving are potential experiments that can demonstrate that the expressed ICAM-1 is also translated into proteins.

We think that there is a misunderstanding as Fig. 3b already shows dose-dependent expression of ICAM-1 at the protein level using flow cytometer.

- In the legend of Fig. 3B, it is indicated that ICAM-1 expression at the protein level was measured by LSR-Fortessa cytometry, but this assay does not appear in the Methods section. Please explain the procedure and clarify whether it was performed in other experiments.

We have added the following information in Methods:

“hSFs at 70-80% confluency were seeded in 24 well plates (Merck 22illipore) in 500µL of DMEM containing 10% EVdepleted FBS and incubated with PvEVs, hEVs (20µg/mL or 60µ/mL), LPS (Sigma) or TNFα (Gibco) (100ng/ml) at 37°C, 5% CO2 for 48 h. hSFs without any stimulation were used as control. Each condition was performed in triplicate. Cells were detached by accutase (Sigma) and stained using CD54- AlexaFlour488 (HCD54, Biolegend) 1/400. Labelled cells were acquired in a LSRFortessa cytometer (BD), and Flow Jo software was used to analyse the data. The population of the cells was selected by FSC-A/SSC-A and cell debris and dead cells were excluded. Afterwards, singlet was selected by FSC-A/FSC-H. Median fluorescence (B530-A) was used for the quantification (Fig. S10). Fold-change ICAM-1 expression was calculated as the ratio between the expression in stimulated cells versus non-stimulated control cells”.

- In order to confirm the translocation of NF-KB to the nucleus, as a complementary approach, I strongly suggest that biochemical nuclear fractionation be performed.

Upon this suggestion, we tried performing these experiments; however, unlike the published literature using 10^6 or 10^7 cells, our experiments in the manuscript were done with 2×10^4 cells. Here using TNFα which readily induces NF-kB nuclear translocation, we tried these experiments with 2×10^5 cells, the highest concentration of cells that would allow us to test this translocation using EVs from the remaining PvEVs pool. Unfortunately, as shown below, using this cell concentration, we were unable to detect nuclear signals in cells treated with TNFα for NF-kB nuclear translocation. Thus, we refrained from performing these experiments with EVs from patients.

However, understanding this reviewers’ concern, we developed an image method to quantify NF-kB nuclear translocation and showed that there is a highly significant difference in this nuclear translocation upon taking up experiments of EVs from patients and healthy volunteers by hSFs. (N=1000 cells, unpaired *t*-test ****p*<0.001) (Fig. S6).

- TO SHOW THAT THE ADHESION INCREASES DUE TO ACTIVATION OF NF-KB IN FIGURE 4A, IT WOULD BE NICE IF THE AUTHORS USE IMMUNOFLUORESCENCE TOGETHER WITH THE LOCALIZATION OF NF-KB.

This is a very nice suggestion. However, all slides prepared in the binding/inhibition assays were fixed with methanol and stained with Giemsa. In trying to address this suggestion, we destained the slides with 100% methanol for five min and ran IFA assays. Unfortunately, using this method, antibodies against NF-kB or against the parasite (PvMSP1) did not show any reactivity (see below); however, the DAPI staining showed clear nuclei including a *P. vivax* schizont with several nuclei. The redish background represents autofluorescence.

- THE SAME EXPERIMENTS FOR CHECKING ICAM-1 AT THE PROTEIN LEVEL ARE SUGGESTED TO DEMONSTRATE THAT ICAM-1 INDEED MEDIATES THE BINDING OF IRBC.

- IN ORDER TO PROVE THAT THE INCREASED ADHESION IS DUE TO ENHANCED NF-KB-DEPENDENT ICAM-1 EXPRESSION, THE EXPERIMENT IN 4A SHOULD BE REPEATED IN THE PRESENCE OF ICAM-1 AND NF-KB INHIBITORS (E.G., SELLECK BIOCHEM A-205804 AND BAY 11-7082).

Ideally, both inhibitors should be used. However, the complexity and scarcity of materials and parasites (see binding/inhibitory assays in Methods) obliged us to use only one of them. After showing that inhibition of NF-kB nuclear translocation also inhibited protein expression of ICAM-1 at the surface of *hSFs* (new Fig. 3d), we performed these experiments with this inhibitor only. Results fully reinforced our model and we thank this reviewer for suggesting these experiments.

METHODS

- LINE 397: THE IVIS-SPECTRUM SYSTEM WAS USED TO MONITOR THE BIODISTRIBUTION OF LABELLED EVS. PLEASE ELABORATE ON THE CONDITIONS IN WHICH THIS EXPERIMENT WAS PERFORMED.

We have now added the following explanation: *“Harvested organs were imaged in the IVIS-SPECTRUM imaging system (PerkinElmer) with the setting*

(Level=High, Em=840nm, Ex=745nm, Epi-illumination, Exposure time=5 sec). To process the images, the same intensity (Min:=5.60E6, Max=6.4E7) was used in all the experiments. Average radiance was used to quantify the signal of EVs”.

- THE PROTOCOL EXPLAINED IN LINES 379-384 IS NOT COMPLETELY CLEAR. I WOULD SUGGEST REPHRASING THE PROCEDURE SO AS TO IMPROVE THE UNDERSTANDING.

We have rephrased this protocol: “Number and size distribution of particles were determined by Nanoparticle tracking analysis (NTA) in a NanoSight LM10-12 instrument (Malvern Instruments Ltd, Malvern, UK) as previously reported²² using NTA software (version 3.2). Samples were diluted in PBS to obtain the optimum measurement range of 1×10^8 – 1×10^9 particles per mL (20-100 particles per frame). Each measurement was performed in technical triplicates”.

- IN LINE 115, DELETE THE WORD ‘EXTRACELLULAR’ FROM THE TEXT “PROTEINS BELONGING TO EXTRACELLULAR EXOSOMES”.

We understand this comment but this term is the one that automatically appears in the Database for Annotation, Visualization and Integrated Discovery (David 6.8). Regardless, we have now removed extracellular from this term.

- IN LINES 432 AND 433, CONTROLS OF EV ACTIVE UPTAKE ARE MENTIONED. PLEASE SHOW THE RESULTS OF THESE CONTROLS WITHIN THE MANUSCRIPT.

These results were already shown in the previous version and are shown in this revised version in Supplementary Fig. S5. Due to space limitations on Fig. 2, we prefer to leave them as supplementary information.

- LINE 471: PLEASE CHANGE THE WORD “TAGMAN” TO “TAQMAN”.

Changed.

- LINE 523: PLEASE CHANGE THE WORD “TWO-TAILS” TO “TWO-TAILED”.

Changed.

MISSING REFERENCES:

- THE AUTHORS SHOULD INCLUDE THE FOLLOWING KEY REFERENCES:

DEMARTA-GATSI C ET AL, (2019). ‘HISTAMINE RELEASING FACTOR AND ELONGATION FACTOR 1 ALPHA SECRETED VIA MALARIA PARASITES EXTRACELLULAR VESICLES PROMOTE IMMUNE EVASION BY INHIBITING SPECIFIC T CELL RESPONSES’

MANTEL ET AL., (2016). ‘INFECTED ERYTHROCYTE-DERIVED EXTRACELLULAR VESICLES ALTER VASCULAR FUNCTION VIA REGULATORY AGO2-MIRNA COMPLEXES IN MALARIA’.

SISQUELLA ET AL. (2017). ‘MALARIA PARASITE DNA-HARBOURING VESICLES ACTIVATE CYTOSOLIC IMMUNE SENSORS’.

The last two references have now been included. We refrained from including the first one as we have focused the manuscript in human malaria and inter-cellular communication.

REVIEWER #2 (REMARKS TO THE AUTHOR):

TODA ET AL HAVE STUDIED THE INTERACTION BETWEEN PLASMA EXTRACELLULAR VESICLE AND SPLEEN FIBROBLASTS IN RELATION TO PLASMODIUM VIVAX CYTOADHESION AND SEQUESTRATION. THIS IS AN INTERESTING ARTICLE WHICH PROPOSES AN INTERESTING HYPOTHESIS ON THE BIOLOGY AND PATHOGENESIS OF P. VIVAX INFECTION. HOWEVER, ALTHOUGH MOST OF THE EXPERIMENTS ARE PROPERLY DESIGNED AND PERFORMED, THEY LACK ADEQUATE STATISTICAL ANALYSES. IN ADDITION, THERE IS A CERTAIN BIAS IN THE ANALYSIS OF THE DATA AND TO SUPPORT THE AUTHORS' CONCLUSIONS, ADDITIONAL DATA ARE REQUIRED.

SPECIFIC COMMENTS

1. THE AUTHORS USED THE TERM "CONCEALED". THIS MEANS THAT THE PARASITE IS HIDING, HIDING FROM WHAT? IF THE PARASITE CYTOADHERES TO FIBROBLAST (AS SHOWN HERE) AND TO ENDOTHELIAL CELLS (AS DEMONSTRATED BY THE SAME GROUP PREVIOUSLY, CARVALHO ET AL, J INFECT DIS, 2010, 202, 638), WHY NOT USE THE TERM CYTOADHERENCE AND SEQUESTRATION AS PROPOSED BY THE SAME AUTHORS IN A PREVIOUS ARTICLES (LOPEZ ET AL, J INFECT DIS, 2014, 209, 1403). I PROPOSE THAT THEY SHOULD AVOID NEW TERMINOLOGY UNLESS THEY PROVIDE A RATIONALE FOR IT.

We thank this reviewer for this thoughtful comment. Accordingly, we have changed the term concealment for cytoadherence and/or sequestration throughout the text.

2. THE MAJOR LIMITATION OF THIS ARTICLE IS ITS STATISTICAL ANALYSES.

3. THE AUTHORS DID NOT SHOW THAT THEIR DATA SHOW THE NORMAL DISTRIBUTION AND THUS CANNOT USE T-TEST OR ANOVA TEST.

THE AUTHORS SHOULD PERFORM KRUSKAL-WALLIS TEST OR MANN-WHITNEY TESTS WHEN DISPLAYING 4 GROUPS IN THEIR HISTOGRAM PLOTS OR TWO IN THE MAIN GRAPH.

2. We have now used the Mann-Whitney test (i) in the results of variability of the individual samples (Fig. 1a), (ii) in the results of the different uptaken assays (Fig. 2b-c), (iii) in the binding/inhibition assays (Fig. 4a) and (iv) in the difference in number of human proteins (Fig. S1).

3. We applied the Kruskal-Kallis test followed by Dunn's multicomparison test to analyse the data on in vivo distribution. As using this statistical test we only reached significant differences between PvEVs and Controls, we decided to remove statistical values from this Figure. Yet, we showed the individual values of each experiment (Fig. 2a).

4. We used unpaired two-tailed *t*-test in the experiments shown in Figures 1e, 3a, 3b, 3d, S6d and S7b as this same test has been used to obtain *p*-values in experiments with *n*=3 in *Nature Communications* [see for examples: Abdalle et al., 10,3778 Figs1-2 (2019); Zeng, et al., 9, 5395 Fig 1 (2018); Sisquella et al., 8, 1985 Fig 7 (2017)].

5. We have now shown the distribution of the different experiments as dots in all figures. Differences were considered significant when the *p*-value was < 0.05. **p* < 0.05, ***p* < 0.01, ****p* < 0.001, *****p* < 0.0001.

4. THE CLAIM THAT THERE IS TROPISM OF THE EV TO THE SPLEEN IS BASED ON THE EXPERIMENTS DESCRIBED IN FIGURE 2. THIS SYSTEM IS ARTIFICIAL AND IS NOT CONCLUSIVE. IN ANY HETEROLOGOUS SYSTEM, THE LIVER AND SPLEEN ARE THE PRIMARY SITES OF FILTRATION. AT LEAST A CONTROL WITH EV FROM INFECTED MICE SHOULD BE USED. THIS SYSTEM SHOULD BE VALIDATED BEFORE ANY CONCLUSIONS CAN BE DRAWN.

We agree with this comment as we have already discussed in the original submission that the spleen and the liver represent organs of the mononuclear phagocytic system where EVs are physiologically taken up. Yet, we showed that there is more than 3-fold uptake by these cells as compared to the uptake of EVs from healthy individuals or from plasma of cows infected with *Fasciola hepatica*. Nevertheless, following this suggestion we have performed an experiment of the in vivo distribution of mice injected with EVs from *P. yoelii* and demonstrated similar tropisms (Fig. S3) (PyEVs = *Plasmodium yoelii* reticulocyte-derived EVs; mEVs = healthy mouse plasma-derived EVs).

5. THE AUTHORS FOCUS ALL THE REMAINING ON THE EFFECT IN THE EV ON HUMAN SPLEEN FIBROBLAST. THIS IS A STRONGLY BIASED APPROACH. THEY DO NOT RULE OUT THE CYTOADHERENCE TO ENDOTHELIAL OR TO MONOCYTES/MACROPHAGES, OR PHAGOCYTOSIS BY MONOCYTE/MACROPHAGES (SEE CHAMBERLAIN ET AL, 2019 STEM CELLS, 37, 652) MAY BE MORE RELEVANT FOR SEQUESTRATION OF THE PARASITE IN THE SPLEEN. THEY SHOULD PERFORM EXPERIMENTS USING ENDOTHELIAL CELLS AND MONOCYTES/MACROPHAGES.

This work is based on a novel working hypothesis in malaria, namely that circulating EVs act as intercellular communicators with human spleen fibroblast facilitating cryptic infections, originated from our own observations on spleen cytoadherence in a rodent malaria model for *P. vivax* ³. Therefore, it is a focused working-hypothesis-based approach using human spleen fibroblasts obtained from transplantation donors. Yet, to address this concern we performed experiments with human spleen endothelial cells and show unequivocally that

these cells readily translocate NF- κ B to the nucleus under normal in vitro growing conditions and that this translocation does not increase ICAM-1 expression (Fig. S6). Based on these results, we refrained from performing binding experiments. Yet, we thank this reviewer for suggesting these experiments as they gave stronger support to the specificity of the signaling mechanism carried by circulating EVs from human patients.

With regard to the suggestion of performing *P. vivax*-binding assays with monocytes/macrophages, as recently reviewed in malaria ⁴, monocytes remain in peripheral blood for only two days before tissue homing to differentiate into macrophages. Moreover, in the spleen, splenic macrophages actively phagocytose malaria-infected red blood cells through interactions with the phosphatidylserine receptor. Last, depletion of macrophages increase the presence of circulating iRBCs facilitating endothelial sequestration. Thus, we believe that under normal physiological conditions monocytes/macrophages are not the natural cells for cytoadhesion but for destruction of malaria iRBCs. As binding experiments in *P. vivax* are extremely difficult to perform (see Methods), we kindly request from this reviewer to understand why we refrained from performing binding experiments with these cells.

6. THE CLAIM, LINE 253, THAT THERE IS NO ENDOTHELIUM IN THE BONE MARROW AND IN THE SPLEEN IS INACCURATE. FOR THE BONE MARROW: TAVASSOLI, PROG CLIN BIOL RES, 1981; RAFFI ET AL., 1995, 86,9, 3353; FOR THE SPLEEN, SEE: QIU ET AL, BLOOD ADVANCES, 2,1130.

We certainly agree with this comment and believe that our original wording was not clear enough as we meant “a region of the spleen not lined with endothelium”. However, to avoid this confounding factor, we have removed this sentence throughout the manuscript and replaced it by spleen microvasculature.

1. de Menezes-Neto A, et al. Size-exclusion chromatography as a stand-alone methodology identifies novel markers in mass spectrometry analyses of plasma-derived vesicles from healthy individuals. *Journal of extracellular vesicles* 4, 27378 (2015).
2. Wiklander OP, et al. Extracellular vesicle in vivo biodistribution is determined by cell source, route of administration and targeting. *Journal of extracellular vesicles* 4, 26316 (2015).
3. Martin-Jaular L, et al. Strain-specific spleen remodelling in Plasmodium yoelii infections in Balb/c mice facilitates adherence and spleen macrophage-clearance escape. *Cellular microbiology* 13, 109-122 (2011).
4. Ozarslan N, Robinson JF, Gaw SL. Circulating Monocytes, Tissue Macrophages, and Malaria. *J Trop Med* 2019, 3720838 (2019).

Reviewers' Comments:

Reviewer #1:

Remarks to the Author:

The manuscript has improved significantly after the revision. I believe that the additional experiments done, as suggested in the revision, support the main conclusions of the paper and add great value to the field of malaria. Nevertheless, before accepting the paper for publication, the following minor comments need to be addressed:

- Use of immunocompetent mice: the results shown in Fig S2A and S2B demonstrate that PvEV uptake in immunodeficient mice follows the same trend as immunocompetent mice. However, there is a major difference in the signal of uptake obtained between immunodeficient and immunocompetent mice. Additionally, the uptake of PvEV, FhEV and hEV in liver is very similar. Is the uptake of PvEVs statistically different to the uptake of FhEVs or hEVs? Please provide the statistics and comment on this issue.
- Table S1: Throughout the manuscript, it is indicated that ten clinical samples were used for the study. However, 13 samples (ten from Brazil and three from Colombia) are specified in Table S1. Given that the parasitemia levels of the Colombian samples significantly differ by an order of almost 100 from the Brazilian ones, please specify which samples were used in the experiments.
- Fig 1D, NTA analysis of particles: The concentration is 1010. Thus, it is not clear whether the authors diluted the sample and obtained a similar concentration to the original measurements. Please clarify.
- Fig 2A, stained PBS for biodistribution assays: If the authors used stained media as a control instead of stained PBS, this should be indicated in the legend in Line 849 and in the Methods section in line 411.
- The authors changed the amount of micrograms of EVs that were used in the biodistribution assay to the amount of particles/ml. This quantitation should be updated also in Lines 421.
- Since this work is mice-related and in the field of malaria EVs, the authors should cite also the third paper we previously asked for.

Reviewer #2:

Remarks to the Author:

The authors have improved the manuscript by answering most of the questions raised by the reviewer.

The statistical analysis is still inaccurate for some experiments. The authors must demonstrate normal distribution of their data, when they use parametric test (t-test). The argument that t-test has been used before is not valid.

Reviewers' comments:

Reviewer #1 (Remarks to the Author):

The manuscript has improved significantly after the revision. I believe that the additional experiments done, as suggested in the revision, support the main conclusions of the paper and add great value to the field of malaria. Nevertheless, before accepting the paper for publication, the following minor comments need to be addressed:

- Use of immunocompetent mice: the results shown in Fig S2A and S2B demonstrate that PvEV uptake in immunodeficient mice follows the same trend as immunocompetent mice. However, there is a major difference in the signal of uptake obtained between immunodeficient and immunocompetent mice. Additionally, the uptake of PvEV, FhEV and hEV in liver is very similar. Is the uptake of PvEVs statistically different to the uptake of FhEVs or hEVs? Please provide the statistics and comment on this issue.

Designing in vivo distribution experiments of EVs is still a major challenge since no general consensus is presently available (Betzer et al., *WIREs Nanomed Nanobiotechnol.*, 2019;e1594). Yet, the use of immunocompetent mice for these experiments is widely accepted (Wiklander et al., 2015 *J Extracell Ves*). Noticeably, the new statistics on the in vivo distribution of EVs from patients and human volunteers presented in the revised version of this manuscript (Fig 2), showed a tendency but not significance with regard to the tropisms for the liver and spleen of EVs from vivax malaria patients in immunocompetent mice. Therefore, as written in the text, we performed a single experiment on biodistribution of these EVs in a nude mouse to demonstrate similar tropisms. The differences in the in vivo distribution seen in the spleen are likely reflecting the more efficient destruction of circulating EVs in the slow flow compartment of immunocompetent animals as compared to immunosuppressed animals. To apply statistics, however, we will need to repeat these experiments with more animals, time and expense. Moreover, due to the new ethical RRR principles of animal experimentation, we will need to fully justify the use of these animals. As these new experiments are unlikely to provide new meaningful insights into the EV-physiological role of circulating EVs in natural malaria infections, we kindly request from this reviewer to consider waiving us from performing them.

- Table S1: Throughout the manuscript, it is indicated that ten clinical samples were used for the study. However, 13 samples (ten from Brazil and three from Colombia) are specified in Table S1. Given that the parasitemia levels of the Colombian samples significantly differ by an order of almost 100 from the Brazilian ones, please specify which samples were used in the experiments.

We apologize for this oversight from our end. The Colombian samples were only used for the binding experiments. We have now clarified this in the new text in line 259:

“To determine if the uptake of PvEVs by hSFs could have a role in cytoadherence of P. vivax-iRBCs, we performed in vitro binding experiments using parasite iRBCs isolated from the blood of P. vivax patients. Of note, in addition to the SEC pool of P. vivax patients from Brazil, we also included another SEC pool of three P. vivax patients from Colombia (Table S1)”.

The parasitemia of the samples from Colombia represented parasites/μl and not parasites/200 leukocytes. This has now been corrected in the new Table.

- Fig 1D, NTA analysis of particles: The concentration is 1010. Thus, it is not clear whether the authors diluted the sample and obtained a similar concentration to the original measurements. Please clarify.

We diluted the samples 80 times in PBS (now indicated in the Methods section in line 320) in order to obtain the optimum measurement range of 1×10^8 – 1×10^9 particles per mL (20-100 particles per frame), and then obtained a similar concentration to the original measurements. Here we indicate the details of the measurement."

hEVs	dilution	particles/frame	measured concentration (particles/ml)	Final concentration (particles/ml)
Rep.1	80	36.5	7.20E+08	5.76E+10
Rep.2	80	37.9	7.46E+08	5.97E+10
Rep.3	80	27.6	5.45E+08	4.36E+10
				5.36E+10
PvEVs	dilution	particles/frame	measured concentration	particles/ml
Rep.1	80	62.3	1.23E+09	9.82E+10
Rep.2	80	60.1	1.19E+09	9.48E+10
Rep.3	80	59.9	1.18E+09	9.44E+10
				9.58E+10

- Fig 2A, stained PBS for biodistribution assays: If the authors used stained media as a control instead of stained PBS, this should be indicated in the legend in Line 849 and in the Methods section in line 411.

We have now written stained media in the legend in Line 828 and in the Methods section in line 414 of the new manuscript.

- The authors changed the amount of micrograms of EVs that were used in the biodistribution assay to the amount of particles/ml. This quantitation should be updated also in Lines 421.

This quantitation has been updated in Line 429 of the new version.

- Since this work is mice-related and in the field of malaria EVs, the authors should cite also the third paper we previously asked for.

This reference has now been added to the manuscript.

Reviewer #2 (Remarks to the Author):

The authors have improved the manuscript by answering most of the questions raised by the reviewer.

The statistical analysis is still inaccurate for some experiments. The authors must demonstrate normal distribution of their data, when they use parametric test (t-test). The argument that t-test has been used before is not valid.

We understand this reviewer's concern; yet, it is important to recall that as mentioned in the previous rebuttal letter, due to the amount of peripheral blood that ethically can be withdrawn from patients during acute attacks made unfeasible to perform all the experiments with EVs from individual patients. Therefore, it was necessary to make a pool of all samples. Accordingly, the *t*-test was used to compare measures which source of variability is not biological, demographic or temporal but inherent technical error, and we assumed normality for the distribution of those experimental replicates. On the other hand, it is known that the *t*-test is robust against deviations from normality if the actual distribution is symmetric, and our data don't show signs of kurtosis or presence of outliers. We have given this explanation in Statistical Analysis:

"All data were analyzed using GraphPad Prism software (Version 8.3.0) and are presented as mean \pm standard deviation. As the source of variability presented in experiments in Figures 1e, Fig. 3a, b, d, Fig. S6d, and Fig. S7b was not biological, demographic or temporal but inherent technical error, P-values were assessed using unpaired and two-tailed t-test assuming normal distributions. In fact, our data didn't show signs of kurtosis or presence of outliers".

Ideally, a large number of replicates would be needed to show normal distribution that allows the canonical use of a parametric *t*-test. However, working with this pool of EV-fractions obtained from malaria patients during their acute attacks precludes the possibility of generating such a large number of replicates. The three replicates shown in our experiments are extremely similar between them suggesting a normal distribution. Therefore, we trust we can maintain the statistical analysis in its current format.

We would like to thank again the reviewers for all of these valuable suggestions which have significantly increase the quality of this manuscript, and hope that it is now acceptable for publication in ***Nature Communications***.

Reviewers' Comments:

Reviewer #1:

Remarks to the Author:

The authors have addressed our comments in a satisfying manner and I have no additional comments.

I believe that this work is now well-deserved to be published in Nature Communications.

Reviewer #2:

Remarks to the Author:

Most of the reviewers' queries are been answered positively However, the statistical analysis is still not adequate and the arguments provided not convincing.

REVIEWERS' COMMENTS:

Reviewer #1 (Remarks to the Author):

The authors have addressed our comments in a satisfying manner and I have no additional comments.

I believe that this work is now well-deserved to be published in Nature Communications.

WE THANK THIS REVIEWER FOR THIS FINAL DECISION

Reviewer #2 (Remarks to the Author):

Most of the reviewers' queries are been answered positively However, the statistical analysis is still not adequate and the arguments provided not convincing.

GIVEN THIS REMAINING CONCERN, WE HAVE DESCRIBED STATISTICS AND SAMPLE TYPE (E.G. POOLING OF SAMPLES AND TECHNICAL REPLICATES) CLEARLY IN METHODS AND FIGURE LEGENDS.

We would like to thank again the reviewers for all their valuable suggestions throughout this reviewing process. They have significantly increased the quality of this manuscript, and hope that it is now acceptable for publication in ***Nature Communications***.

With kind regards, on behalf of all co-authors